# Distinctive features of lincRNA gene expression suggest widespread RNA-independent functions

Alex C Tuck[1,3] , Kedar Nath Natarajan[3,4,6,*], Greggory M Rice[5,*], Jason Borawski[5], Fabio Mohn[1], Aneliya Rankova[1,2], Matyas Flemr[1], Alice Wenger[1], Razvan Nutiu[5], Sarah Teichmann[3,4] , Marc Bühler[1,2]

**Eukaryotic genomes produce RNAs lacking protein-coding potential, with enigmatic roles. We integrated three approaches to study large intervening noncoding RNA (lincRNA) gene functions. First, we profiled mouse embryonic stem cells and neural precursor cells at single-cell resolution, revealing lincRNAs expressed in specific cell types, cell subpopulations, or cell cycle stages. Second, we assembled a transcriptome-wide atlas of nuclear lincRNA degradation by identifying targets of the exosome cofactor Mtr4. Third, we developed a reversible depletion system to separate the role of a lincRNA gene from that of its RNA. Our approach distinguished lincRNA loci functioning in *trans* from those modulating local gene expression. Some genes express stable and/or abundant lincRNAs in single cells, but many prematurely terminate transcription and produce lincRNAs rapidly degraded by the nuclear exosome. This suggests that besides RNA-dependent functions, lincRNA loci act as DNA elements or through transcription. Our integrative approach helps distinguish these mechanisms.**

## Introduction

Eukaryotic genomes are pervasively transcribed by RNA polymerase II (Pol II), producing many long non-protein-coding RNAs (lncRNAs) in addition to mRNAs (Kapranov et al, 2002). LncRNAs are classified by their genomic origins, which include independent transcription units (large intervening noncoding RNAs [lincRNAs]) (Guttman et al, 2009), regions upstream of protein-coding genes (promoter upstream transcripts [PROMPTs] [Preker et al, 2008]) and enhancers (enhancer RNAs).

The biological significance of lncRNAs is strongly debated (Palazzo & Lee, 2015; Deveson et al, 2017), with key questions (i) how many lncRNAs are functionally relevant, (ii) what are the activities of lncRNAs, and (iii) what are the underlying mechanisms? Reported lncRNA functions include many cases where the transcript itself is important (e.g., Xist or Fendrr [Grote et al, 2013; Chu et al, 2015]) and some cases where the RNA product is superfluous, but the act of transcription (e.g., *Airn* [Latos et al, 2012]) or the underlying DNA element (e.g., *Bendr* or *Lockd* [Engreitz et al, 2016; Paralkar et al, 2016]) affects local gene expression.

Of the various lncRNA classes, lincRNAs have most properties in common with mRNAs, including a 5′ m7G cap, poly(A) tail and regulation by key transcription factors (Guttman et al, 2009). As lincRNAs are enriched in the nucleus (relative to mRNAs) (Engreitz et al, 2016), they are primarily suggested to regulate gene expression. This regulation might occur in *cis* (involving adjacent genomic loci) or in *trans* (involving distant, unlinked target genes). LincRNAs are highly differentially expressed between cell types (Cabili et al, 2011) and many have been shown to help specify cell type by acting as functional RNAs (Guttman et al, 2009; Grote et al, 2013; Lin et al, 2014; Leucci et al, 2016). On the other hand, some lincRNA genes could function as DNA elements or via transcription without the need for RNA itself (Engreitz et al, 2016; Ard et al, 2017; Joung et al, 2017). In support of this, lincRNAs are less efficiently spliced than mRNAs and differ in some aspects of 3′ end formation (Melé et al, 2017; Schlackow et al, 2017). Furthermore, some reports suggest that lincRNAs have half-lives similar to mRNAs and are highly expressed in individual "jackpot" cells, whereas others conclude that lincRNAs are less stable and ubiquitously lowly expressed, fuelling the debate of whether the RNA itself is functional (Cabili et al, 2015; Liu et al, 2016; Melé et al, 2017; Schlackow et al, 2017). New approaches must, therefore, identify which lincRNA genes are functionally important and distinguish whether they function as DNA elements, by transcription, or via the RNA product (Bassett et al, 2014).

Two broad strategies are currently used to search for functional lincRNA genes. The first makes predictions based on the properties of the gene or the RNA product, including tissue- or cell type–specific expression, co-expression with other genes, evolutionary conservation, subcellular localisation, or RNA processing and stability (Guttman et al,

[1]Friedrich Miescher Institute for Biomedical Research, Basel, Switzerland   [2]University of Basel, Basel, Switzerland   [3]European Molecular Biology Laboratory, European Bioinformatics Institute, Wellcome Trust Genome Campus, Hinxton, Cambridge, UK   [4]Wellcome Trust Sanger Institute, Wellcome Trust Genome Campus, Hinxton, Cambridge, UK   [5]Novartis Institutes for Biomedical Research, Cambridge, MA, USA   [6]Danish Institute of Advanced Study and Functional Genomics and Metabolism Unit, University of Southern Denmark, Denmark

Correspondence: marc.buehler@fmi.ch; alex.tuck@fmi.ch
*Kedar Natarajan and Greggory M Rice contributed equally to this work.

2010; Tuck & Tollervey, 2013; Necsulea et al, 2014; Cabili et al, 2015). The second uses forward genetic screening to identify lincRNA genes important for a particular phenotype, via targeted or large-scale gene deletions (Sauvageau et al, 2013; Zhu et al, 2016), promoter deletions (Engreitz et al, 2016), integrations of poly(A) cassettes to prematurely terminate transcription (Latos et al, 2012; Engreitz et al, 2016), or CRISPR interference or activation to down- or up-regulate transcription (Joung et al, 2017; Liu et al, 2017).

Approaches to identify functional lincRNA genes are rapidly improving in potency and ease of use. For example, direct RNA-targeting approaches using antisense oligonucleotides (ASOs) (Leucci et al, 2016) or CRISPR-Cas13 (Abudayyeh et al, 2017; Cox et al, 2017) can now efficiently knock down lincRNA expression to directly test the role of the RNA product. This was not the case with previous approaches, such as gene deletion or CRISPR interference, that alter DNA sequence and/or transcription and suffer from unpredictable consequences such as the initiation of novel transcripts (Howe et al, 2017). Despite these recent advances, the efficacy of lincRNA knockdowns can be variable, and there may be off-target effects. Further improvements to these methods, and the testing of additional strategies, is therefore necessary to obtain rapid, efficient, long-lived, and reversible systems to directly target lincRNAs.

With the above considerations in mind, we sought to address three key challenges to understanding lincRNA gene functions (Bassett et al, 2014): (i) to predict which lincRNA genes might be functionally relevant, (ii) to experimentally test whether these genes function in *cis* or in *trans* and whether the RNA product is required for this, and (iii) to more broadly assess the properties of lincRNA transcripts, to help understand how many lincRNA genes produce a functional RNA, versus how many might instead function as DNA elements or via the act of transcription.

To address these aims, we profiled mouse embryonic stem cells (mESCs) and derived neural precursor cells (NPCs) using both single-cell and bulk RNA sequencing. We identified lincRNAs with notable cell type– or subpopulation-specific expression and predicted functionally relevant lincRNA genes. Focusing on 16 of the most promising lincRNA genes, we generated cell lines in which these loci were deleted. In parallel, we developed a self-cleaving ribozyme approach to deplete the expression of some of these lincRNA transcripts, while minimising disruption to the lincRNA gene (i.e., to test the function of the RNA product). Overall, our in-depth analyses of 140 cell lines highlight two key aspects. First, although many lincRNA genes function in *trans*, we predict that up to a third of lincRNA loci act in *cis* to modulate the expression of neighbouring genes. Second, the RNA product may not be required for these local lincRNA gene activities. To investigate this further, we examined lincRNA properties. We profiled the transcriptome-wide direct targets of the nuclear exosome, finding that in contrast to mRNAs, most lincRNAs prematurely terminate transcription (within a few hundred nucleotides of their promoter) and are rapidly targeted for degradation. These properties support recent suggestions that functional lincRNA genes might commonly act as DNA elements or through transcription, without requiring the RNA (Engreitz et al, 2016; Paralkar et al, 2016; Joung et al, 2017) and underscore the importance of directly testing the role of the RNA product.

In summary, our study unravels key differences between lincRNA and mRNA behaviours and provides a comprehensive resource to predict and experimentally test lincRNA gene functions and in particular the role of the RNA product.

## Results

### An atlas of lincRNA expression in mESCs and NPCs at single-cell resolution

To identify lincRNA genes that might contribute to cell identity, we generated an atlas of lincRNA expression across a differentiation time course from mESCs to NPCs using both single-cell and bulk RNA sequencing. We selected this model system because many lincRNAs are expressed and a robust in vitro differentiation protocol exists (Bibel et al, 2004; Wu et al, 2010). Furthermore, a recent screen found that in humans, lincRNA-dependent growth phenotypes are most readily detected in stem cells (Macfarlan et al, 2012), highlighting them as an excellent model system.

We performed strand-specific, rRNA-depleted total RNA-seq for two biological replicates each of mESCs grown in 2i + LIF medium, mESCs grown in serum + LIF (leukemia-inhibitory factor) medium, and during days 3, 6, and 8 of the in vitro NPC differentiation. To obtain a single, comprehensive catalogue of lincRNA genes, we combined a de novo transcript assembly from our data with published lincRNA coordinates (Guttman et al., 2009, 2010; Necsulea et al, 2014) and Ensembl annotations. After filtering the initial set of 7,333 putative lincRNAs for a minimal expression level in at least one of the profiled conditions (>10 reads per kilobase), repeat content (<50%), and no overlap or proximity (<5 kb) to protein-coding genes, we obtained 1,721 high-confidence lincRNAs (Table S1A, "high.quality.biotype" column).

Expression of many lincRNAs (such as *linc1503*) was modulated during neuronal differentiation (Fig 1A and B, and Table S2), suggesting that they are regulated in a cell type–specific manner and might contribute to specifying cell fate. Based on this analysis, we began to shortlist lincRNA genes for functional characterisation in ES cells, considering additional criteria such as robust expression, functional evidence from previous studies (Guttman et al, 2009; Bergmann et al, 2015), and lack of protein-coding capacity based on ribosome profiling data or computational predictions (Chung et al, 2015; Sun et al, 2015) (Tables 1 and S1).

LincRNAs are known to be less expressed than mRNAs (as we also see, Fig 1C), which can either be due to a homogenous low expression or a high abundance in small subpopulations of cells. Furthermore, mESC and NPC cell cultures are heterogeneous, and lincRNAs may be differentially expressed between subpopulations of cells. To explore these issues further, we performed single-cell RNA sequencing of batch-matched cells from the mESC to NPC differentiation using SMART-Seq on the Fluidigm C1 System. After quality control, which 469 cells passed (Fig S1A), variance and clustering analyses on the scRNA-seq data clearly separated the cell states and captured the differentiation trajectory, in good agreement with the bulk RNA-seq and with the expected changes in ESC and NPC markers (Figs 1D and E, and S1C). In addition, subpopulations of mESCs and NPCs expressing specific combinations of genes were apparent (Fig 1E). For example, we clearly resolved three subpopulations of serum + LIF mESCs with different combinations of Oct4 and Nanog

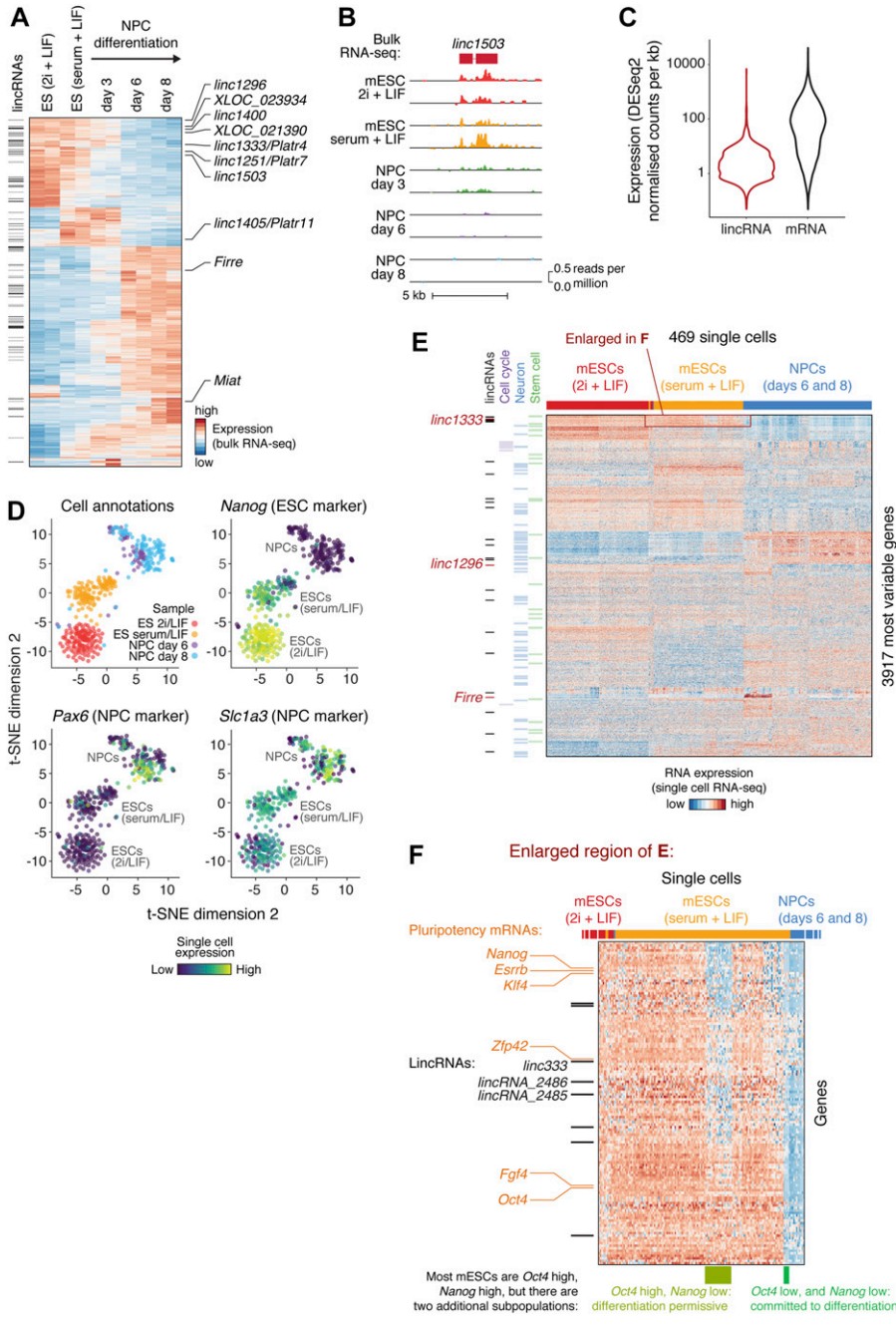

**Figure 1. Single-cell atlas of lincRNA expression during mESC to NPC differentiation.**
**(A)** Bulk RNA-seq heat map of differentially expressed mRNAs and lincRNAs in mESCs grown in 2i + LIF medium, mESCs grown in serum + LIF medium, or derived NPCs. Black bars on the left indicate lincRNAs, and lincRNAs explored in this study are labelled on the right. The 2,000 genes with highest variance were selected. **(B)** An example of a lincRNA with strong differential expression (bulk RNA-seq) during mESC to NPC differentiation. **(C)** LincRNA and mRNA expression distributions from bulk RNA-seq analysis of mESCs (serum + LIF). **(D)** Two-dimensional t-SNE (t-Distributed Stochastic Neighbor Embedding) projection of single cells during mESC to NPC differentiation, coloured by sample ("cell annotations") or the expression of marker genes. **(E)** Heat map showing the most variably expressed mRNAs and lincRNAs during mESC to NPC differentiation (rows), across single cells (columns). Protein-coding genes encoding stem cell, neuronal, or cell cycle factors are highlighted (green, blue, and violet bars), as are lincRNAs (black bars). LincRNAs selected for further study are indicated by red bars. **(F)** Magnified view of the highlighted region (red rectangle) from (E), revealing subpopulations of serum + LIF mESCs with characteristic expression of pluripotency markers (rows indicated with orange labels) and lincRNAs (rows indicated with black bars). Previously reported subpopulations are indicated (columns highlighted by green bars underneath). See also Fig S1 and Table S1A.

expression, corresponding to previously reported "pluripotent," "differentiation permissive," and "committed to differentiation" subpopulations (enlarged in Fig 1F) (Kolodziejczyk et al, 2015). LincRNAs (including *linc1333* and *lincRNA_2485*, which we highlight later) are differentially expressed between these three (and other) subpopulations (listed in Fig S1B). Therefore, the single-cell data reveal an additional layer of complexity in lincRNA expression beyond that seen with bulk RNA-seq analysis.

Remarkably, repeating the single-cell clustering using only lincRNAs (listed in Table S1B) was sufficient to resolve cell types (Fig 2A) and confirmed the lincRNA dynamics detected in the bulk

RNA-seq (Fig 1A). Having verified the quality of the single-cell RNA-seq data, we next addressed the question about heterogeneity of lincRNA expression. To this end, we used two approaches to identify lincRNAs expressed across cell subpopulations. First, we measured cell-to-cell variation for all transcripts by calculating the distance to the median (DM), enabling us to identify highly variably expressed lincRNAs (Fig 2B) (Kolodziejczyk et al, 2015). Second, we mined for individual "jackpot" cells, where lincRNA expression was much higher than the population median (Fig 2C).

We anticipated lincRNAs to be more heterogeneously expressed than mRNAs across single cells (as reported in Liu et al [2016]).

**Table 1.  LincRNA shortlist for functional testing.**

| Gene ID | Translation efficiency | mESC (serum/LIF) expression | NPC (day 8) expression | Reported names | These lincRNAs were selected based on the following criteria |
|---------|------------------------|------------------------------|-------------------------|----------------|--------------------------------------------------------------|
| XLOC_004770 | NA | 129 | 1 | linc1251/Platr7 | ESC specific; reported to be associated with pluripotency |
| XLOC_023375 | NA | 57 | 1 | linc1503 | ESC specific |
| XLOC_046670 | 0.6 | 78 | 5 | linc1400 | ESC specific |
| XLOC_011841 | 0.2 | 1,950 | 90 | linc1283/lncEnc1 | ESC specific; required for ESC colony formation |
| XLOC_017240 | 0.2 | 352 | 31 | linc1296 | ESC specific |
| XLOC_021390 | 0.5 | 117 | 2 | — | ESC specific |
| XLOC_023934 | 0.4 | 97 | 3 | — | ESC specific |
| XLOC_029998 | 0.4 | 493 | 57 | linc1333/Platr4 | ESC specific; reported to be required for pluripotency |
| XLOC_049334 | 0.3 | 132 | 4 | linc1405/Platr11 | ESC specific; reported to be required for pluripotency |
| XLOC_018267 | 0.5 | 141 | 4 | linc1611 | ESC specific |
| XLOC_025367 | 0.1 | 600 | 47 | linc1509 | ESC specific; reported to be required for pluripotency |
| XLOC_009546 | NA | 25 | 1 | Tuna | ESC specific; reported to be required for pluripotency |
| XLOC_051383 | 0.1 | 608 | 728 | linc1409/Firre | Widely expressed; reported to regulate chromatin contacts |
| XLOC_035805 | 0.2 | 97 | 102 | linc1543 | Widely expressed; syntenic with human pluripotency lincRNA |
| XLOC_015936 | NA | 34 | 23 | linc1476/Pvt1 | Widely expressed, with very high levels in some mESCs; oncogene |
| XLOC_036848 | NA | 180 | 4,370 | Miat | Elevated in NPCs; expressed during specific cell cycle stages |

Gene IDs are defined as in Table S1 (from the combined set of published and ab initio lincRNA/mRNA annotations). For translation efficiency, NA = could not be calculated, and for comparison, the median value for mRNAs is 1.0. Expression values in mESCs and derived NPCs were calculated using bulk RNA-seq data, normalising with the DESeq2 size factor function, and averaging the results from the two cell lines we tested (BC8 and CB9). Previously reported names for the lincRNAs are indicated, as well as notable features that we considered when shortlisting these lincRNAs (ESC-specific expression is defined using our RNA-seq data) (Guttman et al, 2009; Hacisuleyman et al, 2014; Jain et al, 2016; Lin et al, 2014; Bergmann et al, 2015).

However, we only observed minor class-wide differences between lincRNAs and mRNAs (Fig 2B and C), which held true across all expression levels. In this respect, our transcriptome-wide results concur with a smaller scale microscopy study of 61 lincRNAs (Cabili et al, 2015). Therefore, in general, the lower average abundance of lincRNAs compared with mRNAs arises from lower expression across all cells, rather than highly subpopulation-restricted expression. That is not to say that lincRNAs are homogenously expressed, but simply that both lincRNAs and mRNAs exhibit similar amounts of heterogeneity. Indeed, we found lincRNAs that were highly variably expressed across single cells (e.g., *Pvt1*, Fig 2D), just as we found lincRNAs that were more uniformly expressed (e.g., *linc1333*, Fig 2E). LincRNAs such as *Pvt1* that accumulate to high levels in "jackpot cells" may function in specific subpopulations of cells.

The variability in lincRNA expression across single cells enabled us to explore in depth the gene regulatory networks within which lincRNA genes might act during differentiation. First, we calculated pairwise correlation coefficients based on single-cell RNA-seq gene expression. We identified ~60,000 significant correlations (Table S3) between 113 lincRNA and ~3,600 protein-coding genes (e.g., *linc1509* correlations, Fig 2F). This provides a rich resource to explore which factors might regulate lincRNA expression and which genes are potentially regulated by lincRNAs. For example, this analysis revealed that *Pvt1* "jackpot cells" also express high levels of *Nmnat2* and *Zic3* (Fig S1D). Intriguingly, these genes are up-regulated in cancers exhibiting high *Pvt1* expression, suggesting that this gene expression module may commonly be up-regulated in highly proliferating cell populations (Cui et al, 2016; Rao et al, 2017). Second, we investigated whether some of the variability in lincRNA levels might arise from cell cycle stage–specific expression. For this, we used the NPC data, as cell cycle stages based on marker expression were more clearly resolved in NPCs (Fig 2G) than in mESCs, which have an unusually short G1 phase. We found examples of lincRNAs specifically expressed during G1/S (e.g., *XLOC_071380*, ~50 kb from the *HoxC* cluster) or G2/M (e.g., *Miat* and *Rmst*) (Fig 2G). Notably, *Miat* was previously implicated in cell cycle regulation (Yi et al, 2017). Therefore, our data reveal lincRNAs expressed in cell subpopulations defined by numerous factors,

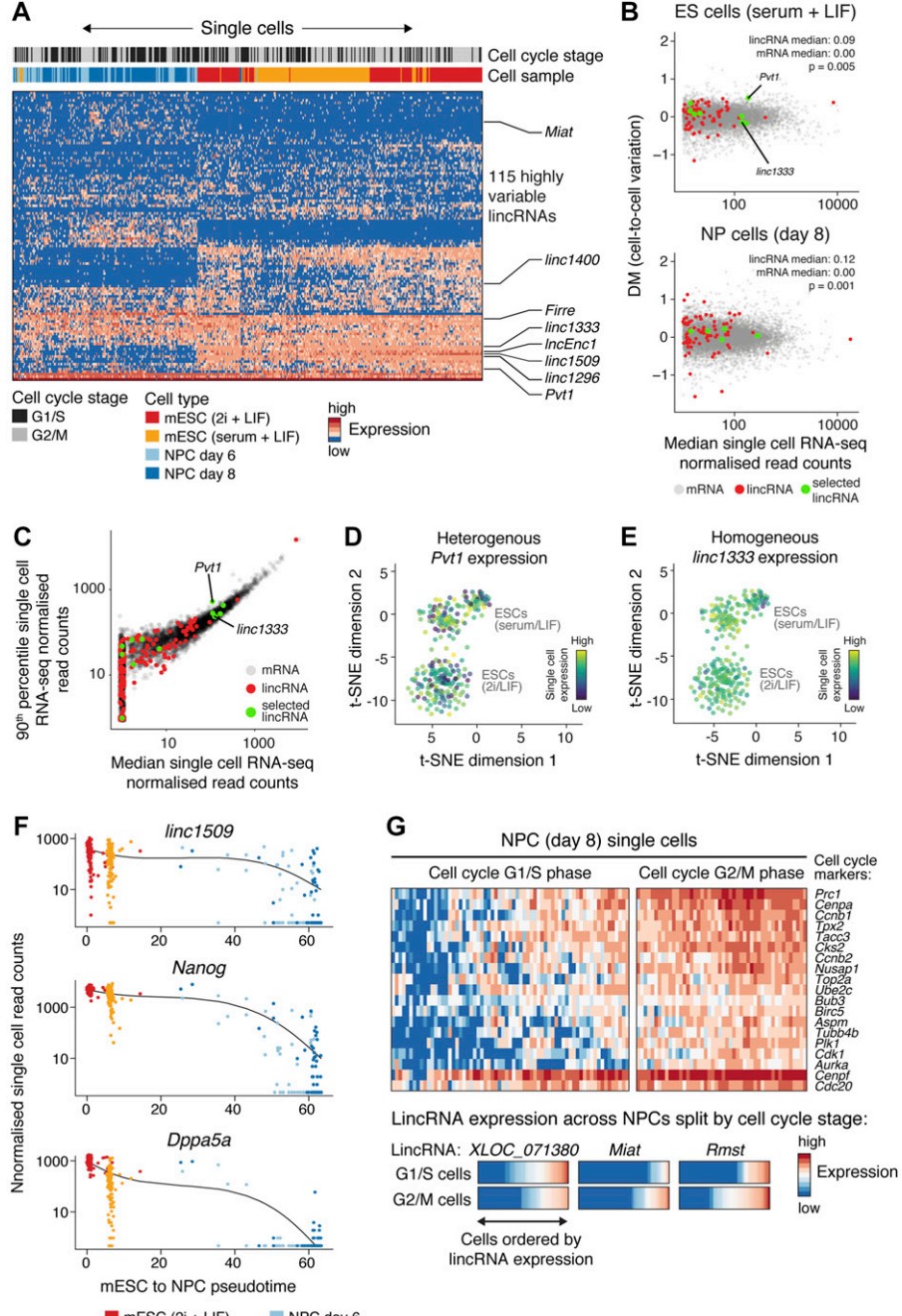

**Figure 2.   LincRNAs expressed in specific subpopulations of mESCs and NPCs.**
**(A)** Expression of highly variable lincRNAs (rows; listed in Table S1B) in single cells (columns), clustered by lincRNA expression. Cell types (mESCs in 2i + LIF or serum + LIF, and day 6 and 8 NPCs) and cell cycle stages (assigned by clustering on cell cycle marker gene expression) are indicated at the top. **(B)** Cell-to-cell variation in expression for mRNAs (grey) and lincRNAs (red), across mESCs (top), or NPCs (bottom), quantified using the DM method (Kolodziejczyk et al, 2015). The median DMs for lincRNAs and mRNAs are listed, and the distributions tested for similarity using the Wilcoxon rank sum test. **(C)** Scatter plot comparing the 90th expression percentile across single cells versus the median expression, for each lincRNA (red) and mRNA (grey). **(D, E)** Single cell expression pattern of a highly variable mESC lincRNA (D, *Pvt1*) and a uniformly expressed mESC lincRNA (E, *linc1333*), overlaid on the t-SNE projection. **(F)** An example of correlated lincRNA-mRNA single cell expression across mESC to NPC differentiation, with cells ordered by pseudotime determined using Monocle (Trapnell et al, 2014). **(G)** Top panels: cell cycle stage assignments for day 8 NPCs based on single cell expression of marker genes (Kolodziejczyk et al, 2015). Bottom panels: examples of three lincRNAs expressed during specific cell cycle stages (*XLOC_071380* = G1/S enriched, and *Miat* and *Rmst* = G2/M enriched). For each lincRNA, the heat bars show expression levels across the 67 G1/S cells (top bar) and 48 G2/M cells (bottom bar). In (B, C), key lincRNAs selected for detailed exploration are highlighted in green. See also Fig S1 and Tables S1A, B, and S3.

including pluripotency and neural markers, and cell cycle stages, hinting at potential functions of these lincRNA genes.

In summary, our integrated bulk and single-cell RNA-seq analysis provides a comprehensive catalogue of 1721 lincRNAs detected during mESC to NPC differentiation (Tables S1 and S2). We classify lincRNAs expressed specifically in mESCs, NPCs, and/or across subpopulations of cells or during specific cell cycle stages and explore the potential of these lincRNA genes to participate in gene regulatory networks (Table S3). Ultimately, however, such computational predictions must be tested. Therefore, based on these analyses, we prioritised 16 lincRNA genes expressed in mESCs that we now sought to investigate experimentally (Table 1).

## Ribozymes: new tools to investigate lincRNAs

We wanted to rigorously test the functions of our shortlisted lincRNA genes (Table 1) and to investigate whether the RNA itself is functional or whether the DNA locus or its transcription is more important (Bassett et al, 2014). To this end, we tested a new method to deplete lincRNA transcripts, without significantly affecting the DNA locus.

Our method is simple and relies only on the stable integration of a 52-nt self-cleaving ribozyme sequence into a lincRNA gene using CRISPR-Cas9. Following transcription, the lincRNA transcript is cleaved and ultimately degraded. Our approach has several advantages: (i) it targets lincRNA transcripts with minimal perturbation to the DNA locus, which can additionally be controlled with an inactive point mutant ribozyme, (ii) it knocks down nuclear targets, (iii) it can be used to study long-duration processes such as cell differentiation, (iv) it does not suffer from nonspecific and off-target effects, and (v) it can be reversed with a blocking ASO to rescue lincRNA expression.

We initially tested two ribozymes, the Hammerhead (HHRz) and Hepatitis Delta virus (HDVRz), for their ability to cleave different reporter constructs (Figs 3A and S2A) (Camblong et al, 2009; Nomura et al, 2013). For the HDVRz, we also tested inducible versions (aptazymes, HDVAz) that fold and cleave upon guanine addition (Nomura et al, 2013), and different HDVAz flanking sequences that suppress background cleavage (without guanine). However, these were less efficient than the HHRz (Fig S2A and B). As a proof of principle, we then knocked down the expression of an endogenous protein-coding gene, G9a, by homozygous integration of the HHRz (Fig 3B). The HHRz reduced G9a expression by up to 81% across four different cell lines (versus five controls), resulting in widespread gene expression changes of known G9a targets (Macfarlan et al,

2012), as measured by bulk RNA-seq (Fig 3C and D). This confirms that the ribozyme approach can be used to identify cases where a gene functions via its RNA product.

We next systematically knocked down the expression of 15 candidate lincRNA genes by genomic integration of the HHRz (guide RNAs are highlighted in red in Table S5), obtaining between two and eight biological replicates for each gene, and assessed knockdown efficiency by RNA-seq (Fig 4A) and in some cases by qPCR (Fig S2C). We identified a range of knockdown efficiencies for lincRNA targets with various stabilities and expression levels (>80% knockdown for linc1405, linc1509, and linc1611) (Fig 4A). The inefficient knockdowns (e.g., of Pvt1, linc1251, and Firre) could be due to the sequence context of the tested site impeding ribozyme folding. Inclusion of a flanking insulator sequence (see the Materials and Methods section) might help for these genes, or an alternative strategy (e.g., ASOs or the HDVRz/Az) could be tested. Importantly, our method can destabilise nuclear transcripts such as linc1405 and linc1509 (Engreitz et al, 2016), supported by a reduction in linc1509 intron (= pre-lincRNA) levels (Fig 4A and B).

A stable knockdown can highlight the impact of lincRNA transcripts on steady-state gene expression and/or across temporal processes (e.g., differentiation or development). However, lincRNAs might perform transient roles affecting cell state and transitions. Furthermore, some lincRNAs exhibit high variability in expression

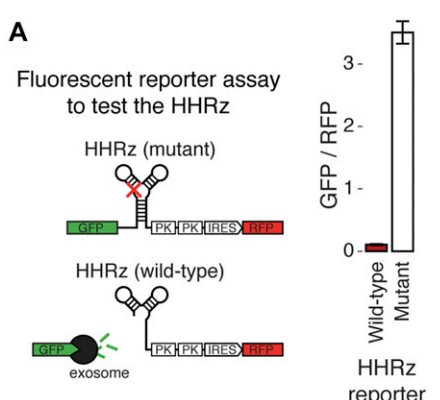

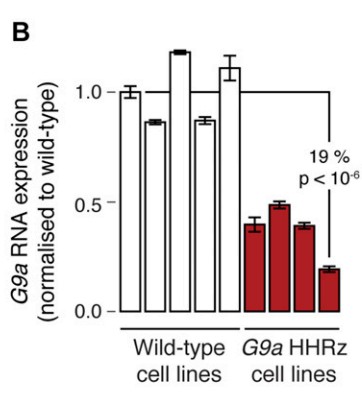

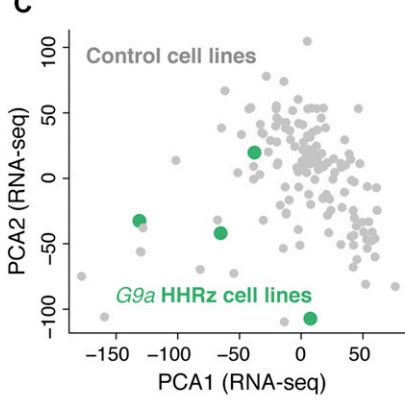

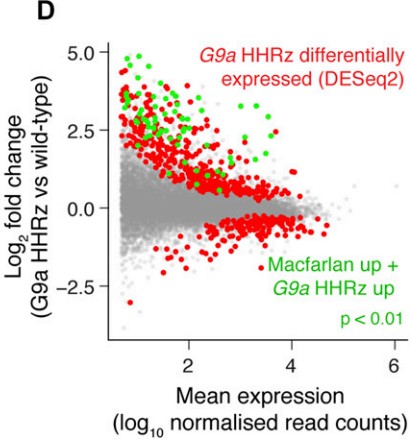

**Figure 3. Development of a ribozyme method to knock down RNA expression.**
**(A)** Plasmid-based fluorescent reporter assay to test RNA knock down by the Hammerhead ribozyme. The HHRz, or an inactive point mutant, was placed between GFP and RFP ORFs, and the relative fluorescence intensity measured (bar chart, right) to quantify cleavage efficiency. Statistical significance was tested using a two-tailed t test, comparing with the wild-type cell line. Mean ± SD is shown (n = 192 technical replicates). **(B)** Assessment of knockdown of endogenous G9a mRNA levels upon genomic integration of an HHRz cassette into the 3′ UTR of G9a. RNA levels were quantified by RT–qPCR and normalised to TBP. Statistical significance was tested using a two-tailed t test, comparing with all wild-type cell lines. Mean ± SD is shown (n = 3 technical replicates). **(C)** PCA to assess the gene expression pattern in four G9a HHRz cell lines (green) relative to 136 cell lines with a wild-type G9a locus (grey). **(D)** Differential expression analysis of G9a HHRz knockdown cell lines, with significantly differentially expressed genes highlighted in red (DESeq2 adjusted P < 0.05), and significantly up-regulated genes also up-regulated in a previous study of G9a knockdown cells highlighted in green (Macfarlan et al, 2012). See also Fig S2.

between clonal cell lines (in part due to their low expression levels), which make comparisons of wild-type and knockdown cell lines challenging and demands many biological replicates. We therefore sought to inhibit HHRz cleavage in our cell lines to rapidly rescue lincRNA levels and thus enable short-lived or subtle lincRNA functions to be studied in an isogenic background.

Initially, we tested a reported HHRz chemical inhibitor, toyocamycin (Yen et al, 2004), to restore levels of *linc1405*. Cells treated with 1.5 µM toyocamycin for 12 h partially recovered linc1405 levels (to ~30% of wild-type), but this was accompanied by massive cell death and perturbation of housekeeping genes such as *TBP* (data not shown). The drastic effect on cellular viability is consistent with toyocamycin effects when used in *Toxoplasma*

*gondii* (Agop-Nersesian et al, 2008). As an alternative strategy, we tested if an ASO designed to bind the HHRz cleavage site could inhibit HHRz activity. In this way, we rescued *linc1405* levels to ~50% of wild-type expression in a dose-dependent manner in 24 h (Fig 4C). Our approach combining the HHRz knockdown and rescue experiments can, therefore, be applied to study both long-lived and short-duration effects of lincRNA expression, in an isogenic background.

In summary, the HHRz approach is a useful addition to the expanding range of methods (e.g., ASOs, CRISPR-Cas13, and shRNAs) for dissecting the functions of lincRNA genes, and in particular, for testing the role of the RNA product (Bassett et al, 2014). Using the HHRz, a reasonable knockdown is achieved for approximately half of the tested lincRNA genes, and in these cases, there are several

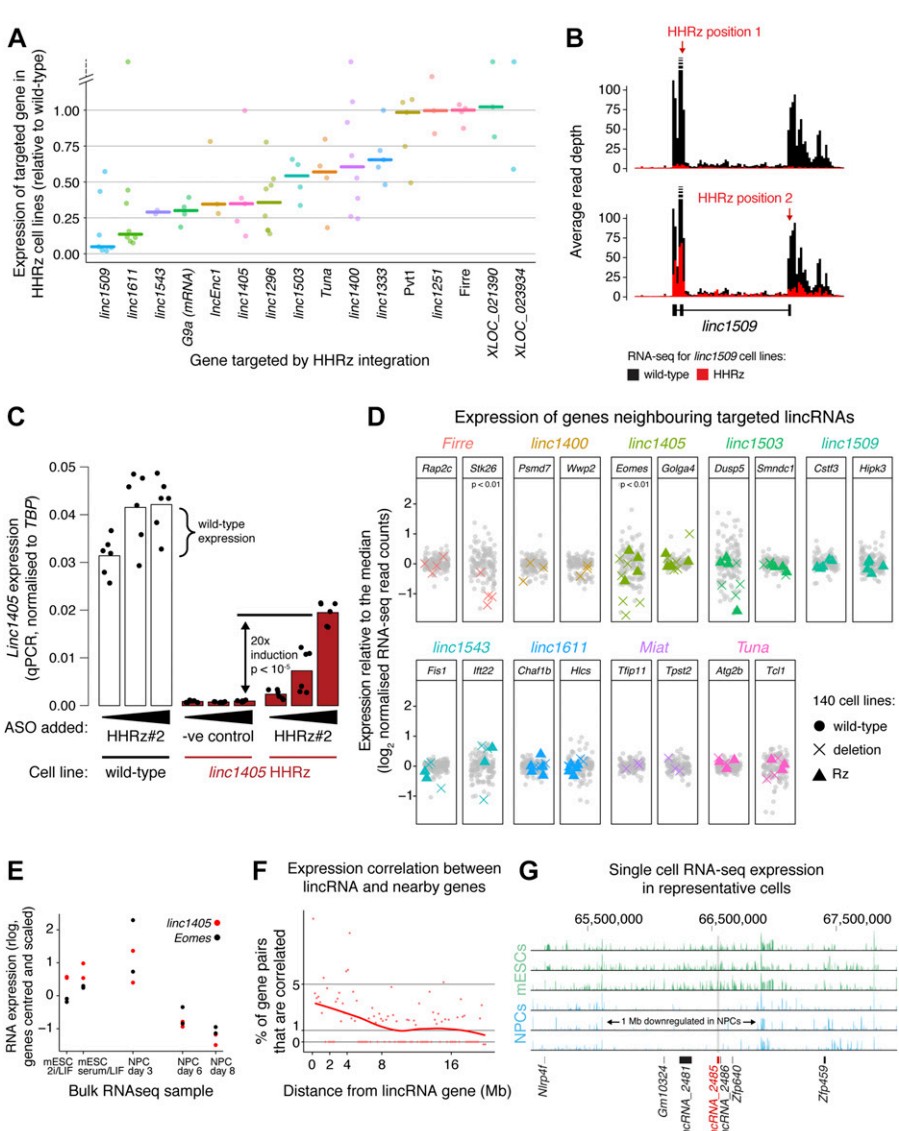

**Figure 4. The ribozyme approach distinguishes RNA-dependent and RNA-independent functions of lincRNA genes.**

**(A)** Diversity of knockdown efficiencies for 15 selected lincRNAs, targeted by genomic integration of the HHRz and assessed using RNA-seq analysis across the collection of 140 cell lines. The expression of the lincRNAs targeted in the knockdown cell lines is shown, normalised to the median of all other cell lines. Each point corresponds to a biological replicate/cell line and the bar is the median across replicates. *G9a* mRNA expression across the *G9a* HHRz cell lines is shown for reference. **(B)** Representative knockdown of *linc1509* using the HHRz. RNA-seq reads across the *linc1509* pre-lincRNA (including introns) are shown, comparing the average read depth in non-targeted cell lines (black) with that in cell lines with the HHRz in *linc1509* exon 2 (top) or exon 3 (bottom) (red). **(C)** Rescue of *linc1405* expression in HHRz depleted *linc1405* cell lines using an oligonucleotide that blocks ribozyme cleavage. *Linc1405* expression measured by qPCR across a wild-type cell line (white) and a *linc1405* HHRz cell line (red), treated with increasing concentrations of the oligonucleotide. As a negative control, the *linc1405* HHRz cell line was treated with a non-complimentary oligonucleotide ("–ve control"). Dots represent individual measurements (two biological replicates, each with three technical replicates), and statistical significance was assessed using a two-tailed *t* test. **(D)** Expression of genes neighbouring eight lincRNA loci, in cell lines where those lincRNA loci were targeted by HHRz integration (coloured triangles) or genomic deletion (coloured crosses). Gene expression was measured by RNA-seq and normalised to median expression in non-targeted cell lines (grey points). Median expression differences between targeted and non-targeted groups of cell lines were tested for significance using a Monte Carlo simulation, applying the Bonferroni correction for multiple hypothesis testing. **(E)** Expression of *linc1405* (red) and the neighbouring gene *Eomes* (black) across the differentiation time course, measured by bulk RNA-seq (n = 2 biological replicates). For each gene, measurements are centred on the mean and scaled by standard deviation. **(F)** Assessment of correlated expression across mESCs and NPCs of *cis*-linked gene pairs involving one lincRNA gene, quantified by Euclidean distance using single-cell RNA-seq data. The % of gene pairs (in 200 kb bins) with Euclidean distances below the 1% significance threshold

(determined in Fig S4A by examining distant gene pairs) is shown. In the absence of proximity-dependent effects, we expect only 1% of gene pairs to be correlated below this threshold. **(G)** Representative single-cell RNA-seq tracks (three mESCs and three NPCs) for the genomic neighbourhood around *linc_2485* (red bar). Genes co-expressed with *linc_2485* (according to Fig S4A) are indicated as black bars. See also Figs S3 and S4 and Table S4A and B.

benefits: there are no off-target effects (as the ribozyme acts in *cis*), the depletion is stable over many cell divisions, and RNA levels can be rescued using a blocking ASO. Therefore, as we now began to examine the functions of our shortlisted lincRNA genes in more detail, where possible, we included lincRNA HHRz knockdown cell lines alongside lincRNA gene deletions.

## Distinguishing *cis* and *trans* functions of lincRNA genes

LincRNAs are primarily suggested to regulate gene expression, either in *cis* (adjacent genes) or in *trans* (distant genes). To examine the functions of our shortlisted lincRNA genes (see Table 1 for main selection criteria), we generated cell lines using CRISPR-Cas9 to delete entire genomic lincRNA loci. A list of cell lines generated, the size of deleted regions, and the Cas9 guides used are provided in Table S4A and B. We analysed these cell lines together with the HHRz cell lines by RNA-seq to identify local and/or global changes in gene expression. The comparison between genomic deletion and ribozyme-mediated knockdown helps to distinguish DNA-/transcription-dependent functions from RNA-dependent functions. In total, we sequenced 140 cell lines (HHRz and genomic deletions, plus wild-type controls, Table S4A) to assess whether the resulting gene expression changes are a specific consequence of lincRNA (gene) perturbation.

We first looked for global transcriptional changes in the lincRNA gene deletion cell lines, using wild-type cell lines and cell lines with neutral integrations or genome edits as controls (Fig S3A and Table S4A). Upon *Miat* deletion, genes involved in cell cycle progression (e.g., *Cul1*, *Cks2*, *Taf10*, and replication-dependent histone genes) or cardiac muscle development (*Sox6*) were misexpressed. This is consistent with the reported roles of *Miat* in myocardial infarction and cell cycle regulation (Ishii et al, 2006; Lai et al, 2017) and with the cell cycle stage–specific expression of *Miat* we observed by single-cell RNA-seq. For the *linc1400* deletion cell lines, the pluripotency factor *Klf2* was down-regulated (Fig S3A), which supports the previously reported role of *linc1400* in influencing the pluripotency network (Guttman et al, 2011). We also observed several changes in the *linc1405* and *Tuna* deletions, and upon *Firre* deletion, the adjacent gene *Stk26* was down-regulated (Fig S3A). Comparing our results with published *Tuna* shRNA, *Firre* ASO and *Firre* deletion datasets (Hacisuleyman et al, 2014; Lin et al, 2014; Bergmann et al, 2015) revealed only a small degree of overlap (Fig S3A), which may be due to differences in the experimental timescales or growth conditions (we used serum + LIF medium, whereas the *Firre* deletion was previously studied using 2i medium), or the conservative nature of our approach whereby we included many negative control cell lines. Nonetheless, for 5/10 lincRNA loci we tested, we observed some evidence of functions in *trans*.

We next looked for local effects in the neighbourhood of deleted lincRNA loci (Fig 4D), finding that deletion of the *linc1405* and *Firre* genes decreased the expression of the neighbouring genes *Eomes* and *Stk26*, respectively. Expression of *linc1405* and *Eomes* are also strongly correlated during mESC to NPC differentiation (Fig 4E, bulk RNA-seq). *Eomes* is 70 kb away from *linc1405*, whereas *Stk26* is 205 kb away from *Firre*. The direct regulation of neighbouring genes by these lincRNA loci suggests that they function as enhancer-like elements. Our results are consistent with an orthogonal study of

*linc1405* function using promoter deletions and poly(A) site insertions and a recent study which identified an intronic region of *Firre* that can function as a DNA enhancer element in a reporter assay (Engreitz et al, 2016; Hacisuleyman et al, 2016). We also observed a tendency for genes adjacent to *linc1543* and *linc1503* to be down-regulated in some but not all cell lines when these lincRNA loci were deleted (Fig 4D). These clonal differences suggest that additional local buffering mechanisms (e.g., epigenetic modifications) maintain neighbouring gene expression and affect the ability of *linc1405* to activate *Eomes*. For the *Miat* deletion, we did not observe changes in neighbouring gene expression, suggesting that the transcriptome-wide changes we see (Fig S3A) arise from *Miat* acting in *trans*.

Extrapolating our results, we predict that up to a third of lincRNA loci potentially influence local gene expression, which would be consistent with estimates from two orthogonal studies (Goff et al, 2015; Engreitz et al, 2016). However, this must be tested further, including in different cell types, and ideally testing hundreds or thousands of loci.

To explore how lincRNA loci might regulate neighbouring genes (i.e., whether the RNA is required), we compared our HHRz and deletion cell lines (Fig 4D). We obtained the clearest data for *linc1405*, for which the locus deletion reduced *Eomes* expression, but the RNA knockdown with the HHRz did not. This suggests that the *linc1405* locus promotes *Eomes* expression by acting as a DNA element or via the act of transcription, and the RNA product is not required. The absence of transcriptome-wide changes in gene expression in the ribozyme datasets (e.g., *linc1611* analysis shown in Fig S3B) confirms that the ribozyme approach does not suffer from off-target effects.

We re-examined all our RNA-seq data and found that numerous lincRNAs were co-expressed with adjacent genes (Figs 4F and S4A and B) and over large linear genomic distances (e.g., in the *lincRNA_2485* neighbourhood, Fig 4G). These broad co-expressed genomic regions suggest that *cis*-regulation involving lincRNA genes could be widespread, although this can also reflect co-regulation of nearby mRNA and lincRNA pairs. To distinguish between these possibilities on a gene-by-gene basis, further lincRNA knockdown and deletion experiments will be required.

In summary, combining our bulk and scRNA-seq data with lincRNA knockdowns and deletions highlights how lincRNA genes can function in a variety of ways (in *cis* or in *trans*, and in an RNA-dependent or RNA-independent manner). This underscores the importance of testing the role of the RNA product with knockdown approaches, such as the HHRz method we introduce, ASOs, or CRISPR-Cas13. These methods can easily show when an RNA product is functional (as we demonstrate for G9a mRNA, Fig 3D). However, as no knockdown method completely removes the RNA, for loci such as *linc1405* that appear to function as DNA elements or via the act of transcription, it is impossible to exclude a role for the transcript without additional evidence. We therefore sought an orthogonal approach to distinguish functional lincRNA transcripts from those that are most likely dispensable by-products.

## Global profiling of the nuclear RNA decay landscape

The functions of yeast lncRNAs are tightly linked to their properties (transcriptional processivity, nuclear stability, and localisation) (Xu et al, 2009; Castelnuovo et al, 2013), and this concept was recently

extended to mammalian lincRNAs (Melé et al, 2017; Schlackow et al, 2017). Therefore, we reasoned that an in-depth analysis of lincRNA properties would shed light on how different lincRNA genes might function. The simplest model is that unstable lncRNAs are by-products of (potentially functional) transcription or DNA elements, whereas lncRNAs with longer, mRNA-like half-lives may accumulate to high levels and function as transcripts. Unfortunately, recent studies of human lincRNA stability reported conflicting results, and so it remains unclear whether lincRNAs are generally less stable than mRNAs or not (Melé et al, 2017; Schlackow et al, 2017).

To resolve this key question, we obtained our own set of transcriptome-wide RNA half-life measurements for mESCs, by monitoring global RNA levels following transcription shut-off by actinomycin D. Compared with mRNAs, lincRNAs had a slightly shorter median half-life when expression levels were considered (Fig 5A and B). Repeating the analysis with more stringent filters for half-life measurements or lincRNA annotations revealed a similar trend, but this was not always statistically significant (data not shown). Furthermore, these analyses do not tell us where or when an RNA is degraded in the cell. Together with the inconclusive results from human cell lines (Melé et al, 2017; Schlackow et al, 2017), this motivated us to develop a more precise way to monitor lincRNA degradation, by directly measuring lincRNA interactions with the decay machinery.

In yeast, lncRNA classes are distinguished by differential targeting by the nuclear exosome and its cofactor complexes (e.g., TRAMP, Trf4/5-Air1/2-Mtr4 polyadenyaltion), and based on exosome knockdown studies, the same appears to be true in mammals (Schlackow et al, 2017). We reasoned that identifying interactions between the nuclear exosome and its substrates would, therefore, offer the deepest insights into lincRNA degradation. To this end, we determined the direct, transcriptome-wide targets of the nuclear exosome–associated helicase Mtr4, using an adapted version of the crosslinking and analysis of cDNAs method (CRAC) (Granneman et al, 2009). Briefly, mESCs with homozygously FLAG-Avi–tagged (Flemr & Buhler, 2015) endogenous Mtr4 were UV irradiated to covalently cross-link protein–RNA complexes, and then the tagged protein was purified. After RNase treatment and denaturing washes, the ~20- to 50-nt RNA fragments directly cross-linked to Mtr4 were sequenced.

This approach offers unique advantages over RNA half-life measurements. First, the cells are not perturbed, and so we accurately capture steady-state exosome substrates. Second, this approach provides a direct readout of nuclear degradation, and so the results are not complicated by the high redundancy between RNA decay pathways. Third, we can capture transient interactions, enabling us to monitor processes such as co-transcriptional decay. Fourth, as Mtr4 is present in various nuclear exosome cofactor complexes (e.g., NEXT and PAXT [Lubas et al, 2011; Meola et al, 2016]), we simultaneously survey many decay pathways. As we obtain high-resolution (~20–50 nt) exosome binding sites, we can distinguish these different decay pathways for individual genes.

We obtained four CRAC datasets for Mtr4, and one for an untagged cell line. As Mtr4 targets specific regions of ribosomal RNA precursors (Lubas et al, 2011), we validated our approach by examining Mtr4-bound pre-rRNA fragments. As expected, this revealed binding in the 5'ETS and 3'-extended 5.8S, with almost no background (Fig S5A). Furthermore, degradation substrates in yeast

are oligoadenylated at the 3' end, providing a landing pad to recruit Mtr4. This appears to be conserved in mammals, although to what extent is unclear (Lubas et al, 2011; Preker et al, 2011). Therefore, we examined our CRAC data for reads with 3' non-genome-encoded oligonucleotide tails, finding a striking enrichment for oligo(A) tails (present on ~10% of Mtr4-bound RNA fragments, Fig 5C). This confirms that we capture bona fide degradation intermediates and establishes oligoadenylation as a major component of mammalian nuclear RNA decay.

We next analysed mRNAs, where Mtr4 acts in multiple decay pathways. We were able to distinguish these decay pathways based on RNA fragments mapping to the following mRNA regions (Fig 5D).

Exons: Mtr4 binding here reflects mature mRNA turnover (Lubas et al, 2011; Meola et al, 2016).

Introns: Mtr4 binding here implicates it in intron removal. Notably, we observed a prominent peak of oligo(A)-tailed reads mapping to intron 3' ends (see also Fig 5E), suggesting that these act as a landing pad for initial Mtr4 recruitment to excised introns.

Promoter-proximal regions: Mtr4 bound abundantly to RNA fragments mapping to the 1-kb region downstream of the transcription start site (TSS) (see also Fig 5F). This reflects the decay of unstable "ptRNAs" arising from early transcription termination in protein-coding genes (Ogami & Manley, 2017).

Upstream antisense regions: Mtr4 also bound abundantly to antisense transcripts produced upstream of the TSS (Fig 5F). This reflects the degradation of very unstable Promoter uPstream Transcripts (PROMPTs) arising from divergent transcription initiation (Preker et al, 2011).

Subsequent analyses revealed that ptRNAs and PROMPTs behave identically and differed only in their orientation relative to the mRNA promoter. For clarity, we therefore refer to these transcripts as sense PROMPTs and antisense PROMPTs (as indicated in Fig 5F).

Our Mtr4 CRAC data, therefore, report on two key stages of the RNA Pol II transcript life cycle—early transcription termination (i.e., transcriptional processivity) and nuclear stability of the mature transcript. Next, we used our data to quantitatively compare these processes for mRNAs and lincRNAs.

## LincRNAs diverge from mRNAs early in transcription

We first compared the nuclear stability of mature lincRNAs and mRNAs by examining Mtr4 CRAC read counts within exons (excluding PROMPT regions) and normalising to Pol II NET-seq as a measure of transcription (performed as described by Schlackow et al [2017]; Fig 5D and F). We used a high-quality list of lincRNA and mRNA 5' end coordinates obtained by filtering for characteristics of active TSSs, such as specific NET-seq and H3K4me3 chromatin immunoprecipitation sequencing (ChIP-seq) signatures (see the Materials and Methods section and column "CRAC.biotype" in Table S1A). We observed that on average, lincRNAs have higher Mtr4 binding across exons than mRNAs (Fig 6A and B), suggesting that mature lincRNAs are more prevalently targeted for nuclear turnover, and in good agreement with their shorter half-lives (Fig 5A and B). Our direct measurements of lincRNA targeting by the nuclear exosome are therefore in good agreement with previous RNA-seq

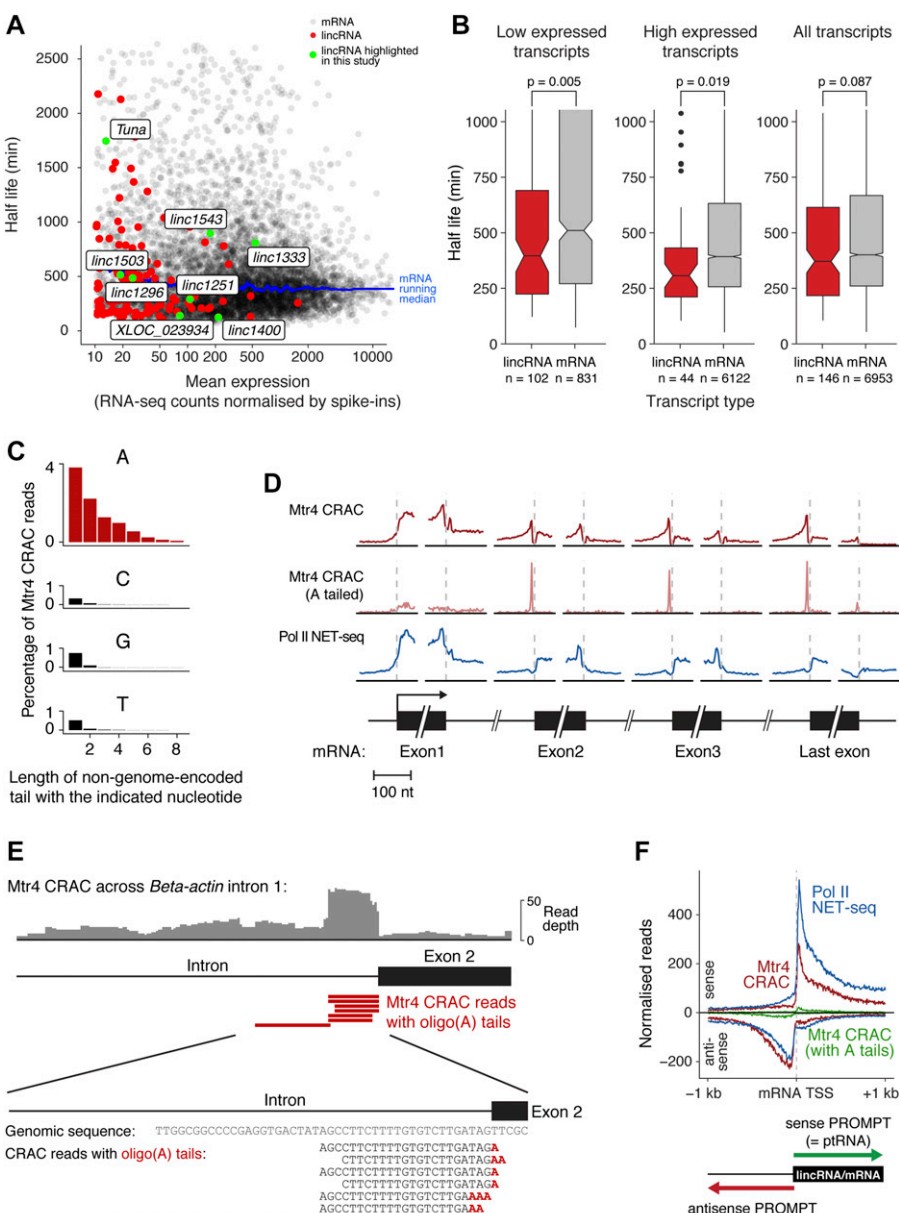

**Figure 5. The nuclear RNA degradation landscape of mESCs.**

**(A)** Half-lives of mRNAs and lincRNAs, calculated from RNA-seq decay curves following actinomycin D–mediated transcription shut-off. Messenger RNAs are shown in black, lincRNAs in red, and lincRNAs highlighted in this study in green. A running median trend line for mRNAs is shown in blue. **(B)** Box plot summary of lincRNA and mRNA half-lives (bar = median; notches = 1.58 × IQR/sqrt(n)), with the distributions tested for similarity using the Wilcoxon rank sum test. The three plots show low expressed (<40 normalised counts), high expressed (>40 normalised counts), or all transcripts (left to right). **(C)** Length of non-genome-encoded oligonucleotide tails detected on RNA fragments captured by Mtr4 CRAC. Oligo(A) tails correspond to degradation intermediates and oligo(C/ G/T) tails are shown as a negative control. **(D)** Metaplots showing the distribution of Pol II Native Elongating Transcript sequencing (NET-seq) (top) and Mtr4 CRAC (middle) RNA fragments around the start and end of mRNA exons and introns (first three and the last exon shown). Mtr4 CRAC RNA fragments with non-genome-encoded 3′ oligo(A) tails, corresponding to initial Mtr4 recruitment sites, are shown separately (bottom). **(E)** An example of Mtr4 CRAC reads mapping near the 3′ end of an intron (β-actin intron 1). Individual sequencing reads corresponding to fragments with 3′ oligo(A) tails are shown as red bars, with an alignment of these sequences below (highlighting in red the non-genome-encoded oligo(A) tails). **(F)** Total distribution of RNA fragments bound to Pol II (NET-seq) or Mtr4 (CRAC) around mRNA TSSs, in the sense (top) and antisense (bottom) directions. "Sense PROMPTs" are defined as short, prematurely terminated transcripts originating from TSSes in the sense direction and "antisense PROMPTs" are similar transcripts that arise in the upstream antisense orientation (see diagram). Mtr4 A-tailed read (as for D) distributions are shown in green. See also Fig S5.

analyses of exosome-depleted cell lines (Schlackow et al, 2017). Importantly, we also noted considerable variability between individual lincRNAs and highlight three examples where this is consistent with their predicted functions (labelled on Fig 6A). Linc1494 had high Mtr4 binding and a short half-life in our actinomycin D experiment, and *linc1494* has previously been suggested to function as a *cis*-regulatory element, based on its conservation with a human enhancer (Engreitz et al, 2016). Our data suggest that the linc1494 transcript is a nonfunctional by-product. Conversely, Platr20 and XLOC_050881 have low Mtr4 binding and long half-lives. *Platr20* expression is associated with the pluripotent state (Bergmann et al, 2015), and *XLOC_050881* is homologous to the human *Nbdy* gene, which encodes a microprotein functioning in nonsense-mediated decay (D'Lima et al, 2016). Our data would be

consistent with *XLOC_050881* and *Platr20* encoding functional RNAs.

In contrast to the relatively moderate differences in Mtr4 binding and half-life measurements for mature transcripts, lincRNAs and mRNAs showed large differences in Mtr4 binding at sense PROMPTs (Fig 6B and C). For protein-coding genes, Mtr4 bound abundantly to antisense PROMPTs but not sense PROMPTs, whereas lincRNA genes had high levels of Mtr4 binding to both sense and antisense PROMPTs. Because Mtr4 binding to PROMPTs reflects early transcription termination (Ogami & Manley, 2017), we conclude that protein-coding genes are more processively transcribed in the forward direction than in the reverse direction. Conversely, transcription from lincRNA TSSs is termination-prone in both directions (Fig 6B and C). Notably, even abundantly expressed lincRNAs had

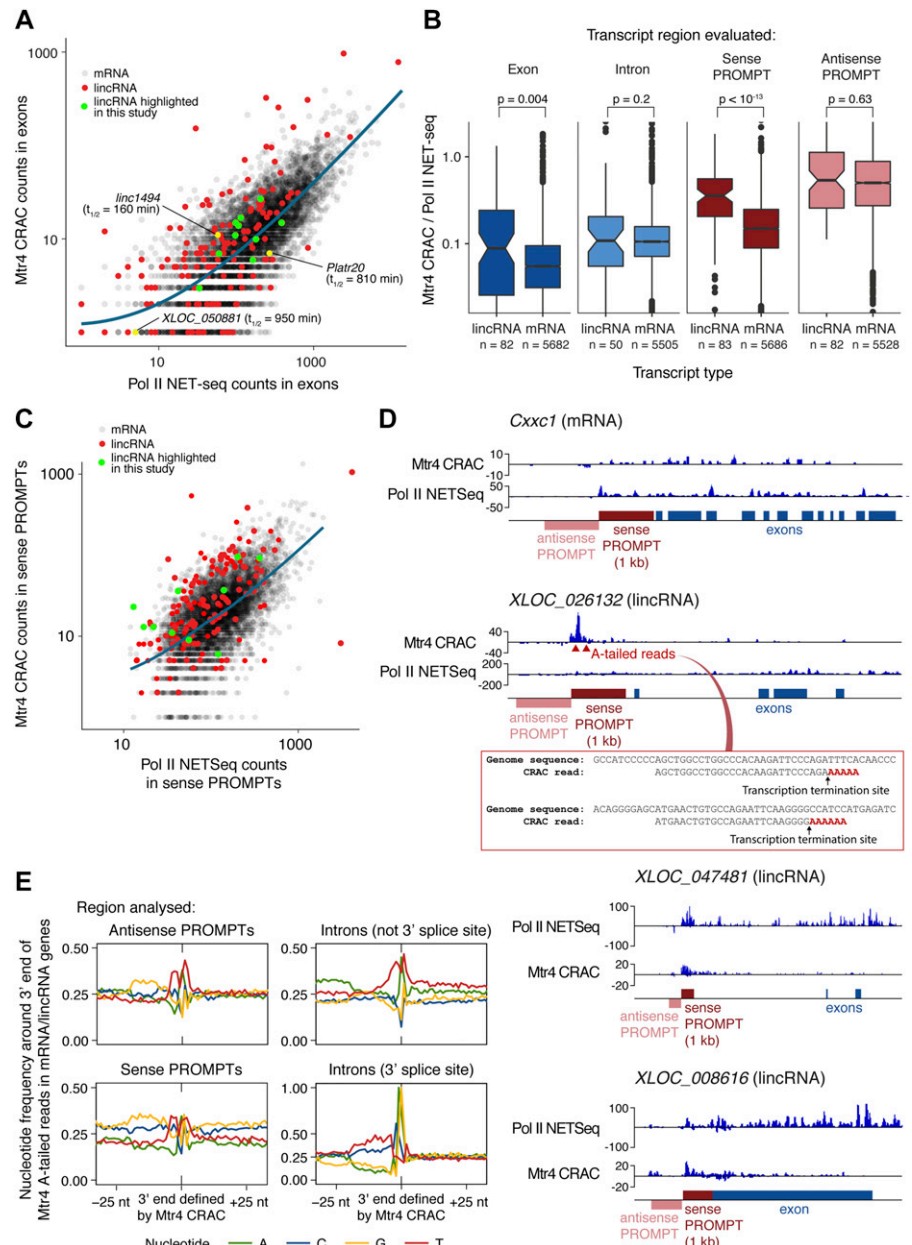

**Figure 6.  Mature lincRNAs are degraded by the exosome and nascent lincRNAs undergo premature transcription termination.**
**(A)** Comparison of Mtr4 CRAC and Pol II NET-seq read counts for mRNA (black) and lincRNA (red) exons, with lincRNAs studied in detail highlighted in green, lincRNAs mentioned in the text labelled, and a trend line for mRNAs shown in blue. For (A–C), only transcripts for which high-confidence TSSes could be assigned are included (see the Materials and Methods section and column "CRAC.biotype" in Table S1A). **(B)** The ratio of Mtr4 CRAC to Pol II NET-seq read counts for mRNA and lincRNA exons, introns, and sense and antisense PROMPTs, using the most stringent set of criteria to select lincRNA genes (genes >2 kb long and >5 kb away from neighbouring coding genes). A Wilcoxon rank sum test was used to compare lincRNAs and mRNAs. The bar indicates the median values, and notches = 1.58 × IQR/sqrt(n). **(C)** Mtr4 CRAC and Pol II NET-seq read counts in mRNA (black) and lincRNA (red) sense PROMPT regions. The CRAC RNA fragments arising here reflect short, early transcription termination products. LincRNAs studied in more detail are highlighted (green) and a trend line for mRNAs is shown in blue. **(D)** Genome browser snapshot showing Mtr4 CRAC and Pol II NET-seq reads across a representative mRNA (Cxxc1; top) and three lincRNAs (XLOC_026132, XLOC_047481, and XLOC_008616). Regions for which reads were counted in are (A–C) indicated, as are oligo(A)-tailed reads that we detect for XLOC_026132, corresponding to early transcription termination sites. **(E)** Genomic nucleotide distributions around PROMPT 3′ ends (= premature transcription termination sites) and intron 3′ ends, defined by analysing Mtr4-bound oligo(A)-tailed fragments that map to lincRNA and mRNA genes. Within introns, two classes of 3′ ends can be distinguished, with nucleotide signatures corresponding to 3′ splice sites (bottom right) or, when intron 3′ ends are filtered out, early termination sites (top right). See also Fig S6.

a high propensity for early termination (Fig 6C). Co-transcriptional cleavage was previously reported across lincRNA genes (Schlackow et al, 2017). Our results support this finding, but suggest that most termination occurs closer to the TSS (within 50–300 nt) than previously appreciated (Fig 5F). This typical behaviour is illustrated in Fig 6D for an mRNA and three lincRNAs.

Having identified elevated early termination as a prominent hallmark of lincRNA transcription, we set out to identify the underlying mechanism. First, we precisely defined transcription termination sites, then we looked for sequence motifs at these sites. To this end, we filtered our Mtr4 CRAC data for reads with 3′ oligo(A) tails, reasoning that as they reveal initial sites of Mtr4 recruitment to excised introns (Fig 5D and E), they would also reveal the 3′ end of premature transcription termination products in sense PROMPT

regions. Indeed, ~8% of sense PROMPT reads possessed oligo(A) tails (Fig 5F), for which we extracted the 3′ genome-encoded co-ordinates (as demonstrated in Fig 6D for lincRNA XLOC_026132). This provided a set of precise Pol II early termination sites in mRNA and lincRNA genes.

Plotting the average nucleotide distribution around these genomic sites revealed an enrichment of Ts on both sides and an enrichment of G further upstream (Fig 6E). This pattern is distinct from that found at the 3′ end of introns and confirms that these are not cryptic splice sites. In yeast, T-rich sequences coincide with the termination site of unstable lncRNAs (Schaughency et al, 2014), and the inherent instability of oligo(dA:rU) DNA–RNA hybrid duplexes can facilitate transcription termination (Martin & Tinoco, 1980). We also found a strong enrichment of Mtr4 CRAC signal immediately

upstream of the 5′ boundary of promoter-proximal nucleosomes in mRNA and lincRNA genes (Fig S6A). This finding is consistent with a recent report that nucleosome remodelling factors can suppress noncoding transcription in mESCs (Hainer et al, 2015). Together, our data point to underlying DNA sequence and nucleosomes as strong driving forces for Pol II early termination, which is prevalent for lincRNA genes.

In summary, we found that lincRNAs are distinguished from mRNAs at multiple steps, including early termination of nascent transcripts as well as exosome-mediated decay of mature lincRNAs, helping to explain the low lincRNA expression levels even in single cells. In contrast to mRNA genes, lincRNA genes generate more short, promoter-proximal RNAs and fewer full-length transcripts. Together with the local enhancer-like functions of lincRNA loci that we experimentally observed, one possibility is that some of these lincRNA genes act as DNA elements or through transcription, with the RNA a dispensable by-product. This is apparently the case for *linc1405* and several other lincRNA loci (Engreitz et al, 2016). However, lincRNAs might also act as transcripts, despite being low in abundance (Seiler et al, 2017). Therefore, the role of the transcript must always be tested experimentally, taking advantage of the rapidly developing range of lincRNA-directed methods.

## Discussion

More than a decade after their discovery, many questions remain unanswered about lincRNA genes, such as (i) which are functionally relevant, and for those that are, (ii) is the RNA itself functional? The greatest challenges in answering these questions are the overwhelming number of lincRNA genes, which must somehow be prioritised for in depth analysis, and the current lack of methods to distinguish RNA-dependent from RNA-independent functions (Bassett et al, 2014). By comprehensively identifying lincRNAs expressed in specific cell types (ESC or NPC), across subpopulations, and during specific cell cycle stages and examining which mRNAs are co-expressed with these lincRNAs, our study provides a systematic and detailed list of candidates for in-depth functional studies. Deleting a selection of these lincRNA loci revealed transcriptome-wide and local regulation of gene expression, suggesting that both modes of lincRNA gene activity are common.

Furthermore, we introduce a ribozyme-based approach that depletes lincRNAs with minimal perturbation of the genomic locus. This can reveal when an RNA product is functional, which we demonstrate using the G9a mRNA as proof of principle. The ribozyme approach was effective for approximately half of the tested lincRNAs, and although we were unable to knockdown two lincRNAs for which we saw evidence of *trans* functions (Miat and Tuna, Fig S3A), orthogonal studies of these lincRNAs support an RNA-mediated role (Lin et al, 2014; Lai et al, 2017). Some of the lincRNA genes we studied in mESCs may have played greater roles in other cell types, as is the case for *Meteor* (*linc1405*), which was recently shown to play a role during mesendoderm differentiation (Alexanian et al, 2017). Although the ribozyme approach requires some effort to implement and does not always achieve a strong knockdown, a major advantage is that the lincRNA depletion is

stable and reversible, so the effects of lincRNA depletion at various stages during mESC differentiation to alternative cell types can be studied. Considered together, ribozyme, ASO, and CRISPR-Cas13 technologies present a powerful range of knockdown approaches that should be widely adopted to test which lincRNA genes function through their RNA product. These approaches would also be suitable for investigating antisense lincRNA–mRNA pairs (the ribozyme is strand specific), which are likely to constitute important regulatory circuits.

Using the ribozyme method, we were able to deplete transcripts by 50–90% from lincRNA loci that appear to act in *cis* as enhancer-like elements (*linc1405*, *linc1503*, and *linc1543*). These knockdowns did not affect neighbouring gene expression, in contrast to the lincRNA locus deletions, and we speculate that these lincRNA loci act independently of the RNA product. As some RNA remains, we cannot rule out the possibility that the remaining RNA was sufficient for function (a caveat of any knockdown method). Parallel approaches can help resolve this ambiguity, such as looking at the properties of the RNA. For example, in the case of the *linc1405* locus, which enhances expression of the neighbouring gene *Eomes*, our CRAC analysis revealed the transcript to be highly unstable and therefore most likely a dispensable by-product. Furthermore, a previous study (Engreitz et al, 2016) reported a direct contact between the *linc1405* and *Eomes* loci and found that blocking *linc1405* transcription with a poly(A) signal does not prevent *linc1405* activating *Eomes* expression. Therefore, the DNA element or promoter-proximal transcriptional activity of the *linc1405* locus appears to be sufficient for contacting and activating expression of *Eomes*. We suggest that a strategy combining locus deletion, poly(A) integration, HHRz/ASO/CRISPR-Cas13 knockdown, and inspection of RNA properties is the most informative approach for investigating new lincRNA genes. Notably, the *Pvt1* locus was also shown to function via its promoter element, which competes for contacts with enhancers (Cho et al, 2018). LincRNA genes that function via modulating chromatin contacts may be a recurring theme.

Genes encoding lincRNAs that function without requiring the RNA product may be common, and recently there has been growing support for this model (Ard et al, 2017). Our comprehensive analyses of lincRNA expression, stability, and transcription provide further evidence for this. Not only are lincRNAs targeted by the exosome for post-transcriptional degradation in the nucleus, in agreement with a recent study (Schlackow et al, 2017), but also more strikingly, we find that lincRNA genes undergo extensive early transcription termination within 1 kb of the TSS. This suggests that lincRNAs are distinguished from mRNAs even earlier in transcription than previously appreciated (Fischl et al, 2017; Schlackow et al, 2017) and that this early stage of transcription is critical for determining RNA fate. Furthermore, this explains why RNA Pol II CTD threonine 4 phosphorylation, a hallmark of transcription termination, is detected immediately downstream of lincRNA promoters (Schlackow et al, 2017).

There is lower Pol II accumulation in promoter-proximal regions of lincRNA genes than mRNA genes, which was previously interpreted as reduced Pol II pausing in lincRNA genes (Schlackow et al, 2017). Instead, our data suggest that the lower Pol II detection at promoter-proximal regions of lincRNAs is indicative of higher

termination and increased Pol II dissociation from the template. By examining the precise 3′ ends of early-terminating lincRNAs and mRNAs, we find a preference for T-rich genomic regions at either side of the termination site and a strong correlation with nucleosome positioning. This suggests that the intrinsic instability of the RNA–DNA duplex in this region, combined with the blocking effect of the nucleosome, promotes Pol II early termination (potentially without requiring canonical cleavage and polyadenylation signals). We speculate that during the early stages of transcription, the Pol II complex is more prone to termination by such mechanisms, whereas the more stable Pol II elongation complex further downstream is resistant. Overall, our data support a model whereby Pol II transcribing mammalian lincRNA genes rarely transitions into productive elongation, as observed in yeast (Milligan et al, 2016; Fischl et al, 2017).

The fact that lincRNAs are low in abundance and prone to premature transcription termination and degradation suggests that in some cases, the lincRNA gene might not require the RNA product for function. However, some lincRNAs (e.g., VELUCT) function despite having a low abundance (Seiler et al, 2017), and low lincRNA levels could be ideal for some RNA-dependent functions. For example, local gene regulation might only require a small number of lincRNA molecules, forming a high local concentration, in which case, the high RNA turnover rate would prevent lincRNAs diffusing away and perturbing other regions. Furthermore, our analyses reveal that some lincRNAs are more stable and/or abundant, which would make them well suited to functioning in *trans*, as appears to be the case for Miat, Rmst, and a growing number of lincRNAs. *Rmst* is particularly interesting, as we find it is expressed preferentially in the G2/M phase in NPCs (Fig 2G), and it was previously reported to recruit Sox2 to the promoters of neurogenic transcription factor genes (Ng et al, 2013). As neural progenitors have a much longer G2/M phase than ESCs, specific expression in G2/M may help restrict *Rmst* expression to differentiating cells.

In summary, it is now clear that when studying a lincRNA gene, it is critical to consider the roles of transcription, the DNA element, and the RNA product (Bassett et al, 2014). This will be greatly aided by the continued development of lincRNA-directed techniques and further exploration of lincRNA properties and their relationship to lincRNA functions.

# Materials and Methods

### Method details

#### Cell lines

For the mESC to NPC differentiation, we used male mESCs from C57BL/6J female × Mus musculus castaneus male (BC8) and Mus musculus castaneus female × C57BL/6J male (CB9) crosses (Sun et al, 2012). For all other experiments, we used male mESCs from a 129 × C57BL/6 cross (Mohn et al, 2008). For tagging endogenous Mtr4, the 129 × C57BL/6 mESCs were modified by the genomic integration of the birA biotin ligase and the CreERT2 recombinase fusion in the Rosa26 locus (Flemr & Buhler, 2015). A list of all cell lines used in this study is provided in Table S4.

#### Cell culture

mESCs were mostly cultured in serum + LIF medium (DMEM, 0.1 mM nonessential amino acids, 1 mM sodium pyruvate, 2 mM L-glutamine, 15% (vol/vol) FBS, 0.1 mM $\beta$-mercaptoethanol, 50 $\mu$g/ml penicillin, 80 $\mu$g/ml streptomycin, and home-made LIF conditioned medium) on gelatin-coated dishes. Where indicated, mESCs were grown in 2i medium (NDiff 227; StemCells, with 0.33 vol/vol LIF-conditioned medium, 1 $\mu$M MEK inhibitor PD 0325901 [Tocris], and 3 $\mu$M GSK3 inhibitor CHIR99021 [Calbiochem]) on gelatin-coated dishes. All cell cultures were grown at 37°C with 5% $CO_2$.

#### ES to NPC differentiation

mESCs were cultured for two passages on gelatin-coated plates before dissociating with trypsin, washing twice in PBS, and seeding ($4 \times 10^6$) on bacteriological Greiner Petri dishes in 15-ml cell aggregate (CA) medium (DMEM with 0.1 mM nonessential amino acids, 2 mM L-glutamine, 10% [vol/vol] FBS, and 0.1 mM $\beta$-mercaptoethanol). The plates were gently shaken twice per day to avoid very large CAs forming. CA medium was replaced after 2 d, by gently transferring the CAs into a 50-ml Falcon tube using a 25-ml pipette, allowing the CAs to settle for 5 min, resuspending the CAs in fresh CA media using a 25 ml pipette, and transferring the CAs into a fresh dish. CA medium was replaced again after a further two days and retinoic acid added (5 $\mu$M final concentration). The medium was again replaced after a further two days, with CA medium + 5 $\mu$M retinoic acid.

For bulk RNA-seq, CAs were washed once in PBS, the supernatant removed, and cells frozen at −80°C. For single-cell RNA-seq, CAs were washed in PBS, dissociated in Accutase, and then passed through a cell strainer to remove cell clumps/debris.

#### Stable cell line generation

Cell lines containing stable hammerhead ribozyme or guanine-responsive Hepatitis Delta virus aptazyme integrations were generated by transfecting SpCas9-sgRNA-2A-mCherry, encoding guides designed to cut close to the integration site (Knuckles et al, 2017), and either an ssODN with 45-nt homology arms (hammerhead ribozyme) or a pBluescript II KS(−) plasmid with the insert and ~500-bp homology arms cloned between KpnI and SacI restriction sites (HDV aptazyme). A corresponding GFP homologous recombination reporter was also co-transfected (Flemr & Buhler, 2015), and after 24 h, GFP + mCherry double positive cells were sorted by FACS and 10,000 seeded on a 10-cm dish. For deletion cell lines, an identical protocol was used, except two SpCas9-sgRNA-2A-mCherry and two pRRE200 reporter plasmids were simultaneously transfected, with the sgRNAs targeting sites 3- to 80-kb apart to delete the enclosed region.

For CRAC, a cell line with Mtr4 tagged with a 3× FLAG-AviTag was generated by transfecting plasmids encoding a Transcription Activator-Like Effector Nuclease pair targeting the Mtr4 N terminus and an ssODN with 45-nt homology arms, together with a puromycin homologous recombination reporter (Flemr & Buhler, 2015). Cells were selected for 36 h with puromycin (2 $\mu$g/ml).

For all cell lines, clones were grown for ~5–7 d and then screened by genomic DNA extraction and PCR using the QIAGEN Fast Cycling PCR kit. Homozygous positives were verified by Sanger sequencing across the target locus.

All transfections were performed using 3 $\mu$l Lipofectamine 3000 reagent per $\mu$g DNA in OptiMEM.

**Life Science Alliance**

**Reagents used in this study.**

| | Source | Product ID |
|---|---|---|
| Cell lines | | |
| C57BL/6J × CAST mESCs | From: Sun et al (2012) | N/A |
| CAST × C57BL/6J mESCs | From: Sun et al (2012) | N/A |
| 129 × C57BL/6 mESCs | From: Mohn et al (2008) | N/A |
| Chemicals, peptides, and recombinant proteins | | |
| DMEM | Gibco | 21969-035 |
| Nonessential amino acids | Gibco | 11140035 |
| 100 mM Sodium pyruvate | Gibco | 11360070 |
| 200 mM L-glutamine | Gibco | 25030024 |
| FBS | Gibco | 10270106 |
| $\beta$-Mercaptoethanol | Sigma-Aldrich | M-7522 |
| Gelatin | Sigma-Aldrich | G-1890 |
| Trypsin–EDTA | Gibco | 25300-054 |
| Dulbecco's PBS | Gibco | 14190 |
| Trypsin (TPCK-treated) | Sigma-Aldrich | T8802 |
| Retinoic acid | Sigma-Aldrich | R-2625 |
| GSK3 inhibitor CHIR99021 | Calbiochem | 361559 |
| MEK inhibitor PD 0325901 | Tocris | 4192 |
| NDiff227 medium | Stem Cells | SCS-SF-NB-02 |
| OptiMEM | Gibco | 31985070 |
| Lipofectamine 3000 Transfection kit | Invitrogen | L3000015 |
| cOmplete Protease Inhibitor Cocktail | Roche | 11836145001 |
| Proteinase K | Roche | 3115879001 |
| SuperScript III | Life Technologies | 18080085 |
| Anti-FLAG M2 magnetic beads | Sigma-Aldrich | M8823 |
| 3× FLAG peptide | Sigma-Aldrich | F4799-25MG |
| Dynabead M-280 Streptavidin | Invitrogen | 11206D |
| RNace-It Ribonuclease Cocktail | Agilent | 400720 |
| TSAP Thermosensitive Alkaline Phosphatase | Promega | M9910 |
| RNasin Ribonuclease Inhibitor | Promega | N2115 |
| Recombinant RNasin Ribonuclease Inhibitor | Promega | N2511 |
| miR-cat 33 conversion oligo pack | IDT | Custom order |
| T4 RNA Ligase 1 (ssRNA Ligase) | NEB | M0204L |
| T4 PNK, T4 polynucleotide kinase | NEB | M0201L |
| Hybond-C Extra membrane | GE Healthcare | RPN303E |
| Kodak BioMax MS autoradiography film | Kodak | 8222648 |
| MetaPhor agarose | Lonza | 50180 |
| NuPAGE 4–12% (wt/vol) polyacrylamide Bis-Tris gels | Life Technologies | NP0335 |
| NuPAGE LDS sample buffer 4× | Life Technologies | NP0007 |
| NuPAGE SDS-MOPS running buffer | Life Technologies | NP0001 |
| NuPage transfer buffer | Life Technologies | NP00061 |
| MinElute Gel Extraction kit | Qiagen | 28604 |
| Proteinase K | Roche | 03115836001 |

**Continued**

| RNase H | NEB | M0297L |
|---|---|---|
| TaKaRa long and accurate (LA) Taq | Clontech | RR002M |
| γ32P-ATP 0.5 mCi 18.5 MBq Spec act. >6,000 Ci/mmol | Hartman | SRP-501 |
| RCC RNA Spike-In Mix 1 | Ambion | 4456740 |
| Single Cell RNA Purification kit | Norgen | 51800 |
| Recombinant DNA | | |
| SpCas9-sgRNA-2A-mCherry | Knuckles et al (2017) | |
| pRRE200 homologous recombination reporter | Flemr & Buhler (2015) | |
| pRRP200 homologous recombination reporter | Flemr & Buhler (2015) | |
| pBluescript II KS(−) | Stratagene | |
| psiCHECK-2 | Promega | C8021 |
| Oligonucleotides are listed in Table S5 | IDT | |

### Ribozyme reporter assays

To test the HDV ribozyme and aptazymes, the psiCHECK-2 vector was used, with the various sequences cloned downstream of the *renilla* gene. The plasmids (100 ng) were transfected into mESCs using the Lipofectamine 3000 Transfection kit, and after 24 h of incubation, the medium was replaced with fresh medium containing either guanine (100 μM with 400 μM NaOH) or 400 μM NaOH (control treatment). Luciferase assays were performed 48 h post-transfection using the Dual-Luciferase Reporter Assay System according to the manufacturer's protocol. Luciferase activities were measured with a microplate luminometer (Centro LB 960; Berthold Technologies). Renilla was normalised to firefly and all values represent the mean ± SD from three biological replicates.

To test the hammerhead ribozyme, the wild-type or mutant ribozyme sequences were inserted into the 3′-RNA end accumulation during turnover (TREAT) reporter (Horvathova et al, 2017) downstream of the GFP ORF. U2OS cells were transfected with the HHRz-active or HHRz-inactive reporter in six-well dishes. Selection was performed using G418 at 500 μg/ml in McCoy's 5A supplemented with 10% FBS. After one week, cells were expanded and then sorted by FACS for RFP+ and GFP+ cells. The cells were further expanded and sorted a second time. Fluorescence intensities were measured using the Yokogawa CV7000 high-throughput Cytological Discovery System. The following active and inactive HHRz sequences were used, including flanking regions predicted using the AntaRNA web server (Kleinkauf et al, 2015) and checked for minimal ribozyme-flank interactions using RNAstructure (Reuter & Mathews, 2010).

Active:
CGUAGAUCUCCCCCCCACCCCUAAAGAAUACCACGCCUGUCACCGGAUGU
GUUUUCCGGUCUGAUGAGUCCGUGAGGACGAAACAGGAACACAAAUCAC
CCGCUCUCUAAUACACACAUUCACCUAAUAGAA
Inactive:
CGUAGAUCUCCCCCCCACCCCUAAAGAAUACCACGCCUGUCACCGGAUGU
GUUUUCCGGUCUAAUGAGUCCGUGAGGACGAGACAGGAACACAAAUCAC
CCGCUCUCUAAUACACACAUUCACCUAAUAGAA

### Ribozyme inhibition with an ASO

Transfections of 15 pmol fully modified 2′-O-methoxy ethyl/phosphorothioate oligonucleotides were performed using 1.5 μl Lipofectamine 3000 reagent and 100 μl OptiMEM, which was added to 250,000 mESCs in 250 μl mESC medium. For lower concentrations, less of the oligo/Lipofectamine/OptiMem mix was added (20 μl or 4 μl). The medium was replaced after 6-8 h with fresh mESC medium, and RNA extracted after a total of 24 h using the Norgen Single Cell RNA purification kit. The oligos GAAAACACATCCGGTGACAG (HHRz#2) and CGTTCTCGAGCCGACTCACA (negative control) were used. The antisense oligonucleotides were fully modified with 2′-O-(2-methoxyethyl) chemistry (2′-MOE) and C nucleotides were further modified with a methyl at the 5 position (5-methyl-2′MOE-C).

### qRT–PCR

For qRT–PCR of mESC ribozyme cell lines, RNA was extracted using the Agilent Absolutely RNA Miniprep kit and 500 ng used for reverse transcription with the Primescript RT Kit. qRT–PCR was performed using the SsoAdvanced SYBR Green Supermix on a CFX96 Real-Time PCR System. Relative RNA abundance was calculated from CT values using the ΔCT method and normalising to TBP mRNA levels.

### Sequencing library construction

For single-cell RNA sequencing, 4,000 cells were loaded onto a 10- to 17-μm Fluidigm C1 Single Cell mRNA Seq IFC, and the cells were captured according to the manufacturer's instructions. The captured cells were manually annotated under a microscope, to identify capture chambers containing cell debris, dead cells, no cells, or multiple cells. Reverse transcription and cDNA pre-amplification was then performed in the 10- to 17-μm Fluidigm C1 Single Cell mRNA Seq IFC using the SMARTer Ultra Low RNA kit for the Fluidigm C1 System and Advantage 2 PCR kit. cDNA was harvested and diluted to 0.1–0.3 ng/μl. Sequencing libraries were then prepared using the Nextera XT DNA Sample Preparation kit and Nextera XT DNA Sample Preparation Index kit according to the Fluidigm instructions, "Using the C1 Single-Cell Auto Prep System to Generate mRNA from Single Cells and Libraries for Sequencing." The 96 libraries from one IFC were pooled and 150-bp paired-end sequencing performed on the Illumina HiSeq2000.

For bulk RNA sequencing of mESCs and NPCs from the differentiation time course, RNA was extracted and libraries prepared using the Wellcome Trust Sanger Institute sample preparation

pipeline with Illumina's TruSeq stranded RNA Sample Preparation Kit including ribosomal depletion step (RiboMinus). RNA and library quality control was performed by the WTSI sequencing facility. For bulk RNA sequencing of mESC ribozyme cell lines, RNA was extracted using the Agilent Absolutely RNA Miniprep kit, ribosomal RNA depleted using the ribo-zero rRNA removal kit, and libraries prepared using the ScriptSeq V2 RNA-Seq Library Preparation kit.

### CRAC

mESCs were grown in 2 × 15-cm dishes to ~80% confluency, the dishes washed twice with PBS, the PBS removed, and cells cross-linked on ice (with the dishes facing up) in a Stratagene Stratalinker 2400 (400 mJ·cm$^{-2}$). The cells were lysed by incubating with 10 ml RIPA buffer in the dish on ice for 30 min (50 mM Tris–HCl, pH 7.8, 300 mM NaCl, 1.0% Nonidet P40 substitute, 0.1% SDS, 10% [vol/vol] glycerol, 0.5% sodium deoxycholate, 1 mM $\beta$-mercaptoethanol, and 1× cOmplete Protease Inhibitor Cocktail). The cells were further disrupted using a cell scraper and then lysates collected and centrifuged at 6,500 $g$ for 20 min at 4°C. The supernatant was snap-frozen in liquid nitrogen until required. Note that milder extraction conditions (e.g., 50 mM Tris–HCl, pH 7.8, 150 mM NaCl, 0.5% Nonidet P40 substitute, and 1× cOmplete Protease Inhibitor Cocktail) can be used for cytoplasmic or non-chromatin–associated proteins, which will greatly improve subsequent binding to the FLAG beads.

Lysates were thawed on ice, and incubated with 100 µl anti-FLAG M2 magnetic beads overnight. The supernatant was then removed, and beads washed three times with 1 ml TN150 (50 mM Tris–HCl, pH 7.8, 150 mM NaCl, and 0.1% Nonidet P40 substitute). Protein–RNA complexes were eluted by incubating the beads in 1.5 ml TN150 supplemented with 5 mM $\beta$-mercaptoethanol and 0.3 mg/ml 3× FLAG peptide, rotating at 4°C for 2 h. The eluate was then incubated with 50 µl Dynabeads M-280 Streptavidin, rotating at 4°C overnight. The beads were washed twice in TN600 (50 mM Tris–HCl, pH 7.8, 600 mM NaCl, 0.1% Nonidet P40 substitute, and 5 mM $\beta$-mercaptoethanol) and twice in TN150 supplemented with 5 mM $\beta$-mercaptoethanol. RNA was then fragmented by incubating the beads in 500 µl TN150 supplemented with 5 mM $\beta$-mercaptoethanol and 1 µl of 0.1 U diluted RNace-IT. After 4 min at 37°C, the reaction was quenched by replacing the solution with 400 µl WBI (wash buffer I: 50 mM Tris–HCl, pH 7.8, 300 mM NaCl, 0.1% Nonidet P40 substitute, 5 mM $\beta$-mercaptoethanol, and 4.0 M guanidine hydrochloride). The beads were washed twice more in WBI, then three times in 400 µl 1× PNK (50 mM Tris–HCl, pH 7.8, 10 mM MgCl$_2$, 0.5% Nonidet P40 substitute, and 5 mM $\beta$-mercaptoethanol).

The following four enzymatic reactions were then performed in 80 µl 1× PNK buffer (omitting the Nonidet P40 substitute), to ligate 3' and 5' adapters onto the RNA fragments. After each enzymatic reaction, the beads were washed once in WBI and thrice in 1× PNK.

(i) Alkaline phosphatase treatment (30 min, 37°C): 8 U TSAP and 80 U RNasIN.
(ii) 3' linker ligation (overnight, 16°C): 0.1 nmol miRCat-33 DNA linker, 40 U T4 RNA Ligase 1, 80 U RNasIN, and 12.5% (vol/vol) PEG8000.
(iii) 5' phosphorylation (1 h, 37°C): 40 U T4 PNK and 2 µl γ32P-ATP (after 30 min, add 1 µl 100 mM rATP and an additional 20 U T4 PNK).

(iv) 5' linker ligation (overnight, 16°C): 0.2 nmol 5' linker, 40 U T4 RNA Ligase 1, 1.25 mM rATP, 80 U RNasIN, and 12.5% (vol/vol) PEG8000.

After the final reaction, the beads were washed three times in WBII (50 mM Tris–HCl, pH 7.8, 50 mM NaCl, 0.1% Nonidet P40 substitute, and 5 mM $\beta$-mercaptoethanol), resuspended in 30 µl 1× NuPAGE LDS sample buffer, heated at 95°C for 2 min, and the eluate quickly removed and loaded onto a NuPAGE 4–12% polyacrylamide gel. The gel was run at 100 V for ~1 h, and then protein–RNA complexes transferred to Hybond-C extra nitrocellulose membrane (Amersham) at 150 V for 1.5 h using a wet transfer system and NuPAGE transfer buffer with 15% methanol. The membrane was then briefly dried, exposed to BioMax MS film (4 h to overnight), and the region corresponding to the protein–RNA complex cut out.

The membrane slice was then incubated in 400 µl WBII with 1% (wt/vol) SDS, 5 mM EDTA, and 100 µg Proteinase K at 55°C for 2 h. The solution was then removed to another tube, 50 µl 3M NaAc, pH 5.2, and 500 µl of 1:1 phenol:chloroform mix added, and the mixture vortexed and centrifuged at 14,000 $g$ for 20 min. The top phase was transferred into a new tube and 1 ml ethanol and 20 µg glycogen added. The solution was stored at –20°C overnight to precipitate RNA and then centrifuged at 14,000 $g$ for 30 min. The RNA pellet was washed once with 70% ethanol and allowed to briefly air-dry, before resuspending in 11 µl water + 1 µl 10 µM miRCat-33 RT oligo + 1 µl 10 mM dNTP mix. The solution was heated to 80°C for 3 min, snap-cooled on ice for 5 min, then the following mix added: 4 µl 5× first-strand buffer (SSIII kit) + 1 µl 100 mM DTT (SSIII kit) + 1 µl recombinant RNasIN. After incubating for 3 min at 50°C, 200 U of SuperScript III was added and the reverse transcription allowed to proceed for 1 h at 50°C. The reaction was stopped by heating to 65°C for 15 min, then RNA digested with 10 U RNase H at 37°C for 30 min. PCR reactions (80 µl) were then prepared, each with 2 µl cDNA, 10 pmol P5, 10 pmol PE, 12.5 nmol each dNTP, and 2.5 U LA Taq. Typically, we ran five PCR reactions and then concentrated the products by ethanol precipitation before resolving on a 3% metaphor agarose gel in 0.5× TBE. A smear corresponding to the size of the two adapters plus inserts (total size ~120–300 bp) was then cut out and DNA extracted using the MinElute Gel Extraction kit, eluting in 20 µl water. If the experiment was successful, we repeated the PCRs with the remaining half of the cDNA.

### Half-life measurements

To measure transcriptome-wide RNA half-lives, 300,000 mESCs were seeded per well of 2 six-well dishes and grown for 48 h in serum + LIF medium. The medium was replaced by fresh medium containing 5 µM actinomycin D (from a 5 mg/ml stock in DMSO) and the cells incubated for 30, 90, or 240 min. A mock treatment (240 min) was included, using medium with the same amount of DMSO but no actinomycin D. After the indicated times, the wells were washed twice with 37°C PBS and RNA extracted using the Agilent Absolutely RNA Miniprep kit. ERCC RNA spike-ins were added to the lysis buffer (1.7 µl of a 1:10 dilution per sample) before it was added to the cells. Ribosomal RNA was then depleted with the ribo-zero rRNA removal kit and high-throughput sequencing libraries prepared using the ScriptSeq V2 RNA-Seq Library Preparation kit.

### Quantification and statistical analysis

#### LincRNA annotations

To compile a reference set of lincRNA annotations, features from the following sources were combined:

(i)  Necsulea et al (2014) lincRNA transcripts;
(ii)  Guttman et al (2010) ES and NPC lincRNA transcripts; and
(iii)  Ensembl GRCm38.75 lincRNAs, miscRNAs, processed_transcripts, and antisense features.

Transcripts were excluded if they overlapped in either orientation with protein-coding Ensembl transcripts or a variety of "classical" noncoding RNAs defined by Ensembl (including snoRNAs, miRNAs, rRNA, nonsense-mediated decay transcripts, and pseudogenes). In cases where the remaining lincRNA transcripts overlapped, they were grouped into single clusters ("genes"). A small number of the resulting clusters were very long (>100 kb), corresponding to regions where individual genes could not easily be distinguished, and were excluded from subsequent analyses. The lincRNA annotations were then combined with all Ensembl non-lincRNA annotations to make a single reference file.

For some analyses, this reference file was also supplemented with our own set of lincRNAs assembled from bulk RNA-seq data using Cufflinks. First, reads from 10 samples (2× mESCs in serum + LIF, 2× mESCs in 2i medium, 2× NPCs day 3, 2× NPCs day 6, and 2× NPCs day 8) were mapped to the mm10 genome with Tophat2, including splice sites from the reference file defined in "LincRNA annotations." The 10 alignment files were then provided to Cufflinks along with the Ensembl/Necsulea/Guttman reference file for de novo transcript assembly, resulting in a merged set of transcript annotations. Any novel transcripts that did not overlap with Ensembl genes, or overlapped with a lincRNA gene, were provisionally classed as lincRNAs. This file contains many de novo–defined unannotated transcripts and was used whenever we wanted to quantify expression of the most complete set of noncoding RNAs. However, for analyses where we wanted to focus on a higher quality set of lincRNAs, we applied the following additional filters:

(i)  exonic sequence >1 kb,
(ii)  <50% of the exonic sequence overlaps with RepeatMasker features,
(iii)  putative lincRNAs are >5 kb away from mRNAs or "classical" noncoding RNAs, and
(iv)  expressed at ≥5 DESeq2 normalised counts per kb in at least one mESC or NPC bulk RNA-seq sample.

LincRNAs and mRNAs that pass these filters are indicated in the "high.quality.biotype" column of Table S1.

To define a set of 15 representative lincRNA genes for functional studies, we additionally considered the following filters, as well as extensive visual inspection of our RNA-seq datasets for ~200 possible candidates:

(i)  low translational efficiency defined using riboSeqR analysis of mouse ribosome profiling data (Array Express study E-MTAB-2934),
(ii)  noncoding prediction by lncRScan-SVM (Sun et al, 2015), and

(iii)  expressed at >30 DESeq2 normalised counts per kb in at least one mESC or NPC sample.

Genomic regions with 20 or more genes per 100 kb were also excluded as these were often incompletely assembled genes in regions of lower sequencing coverage.

To quantify lincRNA versus mRNA stability and transcription termination (half-life measurements and CRAC/NET-seq analyses), we filtered the "high.quality.biotype" lincRNAs/mRNAs further to obtain transcripts for which there was good supporting evidence that the annotated 5′ end was accurate (e.g., specific Pol II and H3K4me3 signatures)—see "Calculating Mtr4:PolII ratios for PROMPTs, exons and introns." For these analyses, we did not always require the lincRNA to be 5 kb away from a protein-coding gene, so long as its TSS was the "stronger" one (based on NET-seq signal).

#### Bulk RNA-seq analysis (ES to NPC time course)

Reads for the 10 samples were mapped to the mouse genome (mm10) with Tophat2 and counted for features in the combined reference file (including de novo lincRNAs) using ht-seq count (mode = union). Feature counts were normalised using the rlog function in DESeq2 and the 1,000 genes with the highest variance plotted as a heat map, clustered based on pairwise Euclidean distances between genes.

#### Single-cell RNA-seq analysis (ES to NPC time course)

Paired-end reads were preprocessed using Trimmomatic, with the following settings: ILLUMINACLIP:Nextera_PE_adapters.fa:2:30:10:1: true LEADING:28 TRAILING:28 SLIDINGWINDOW:1:28 MINLEN:50. The reads were then aligned to mm10 using gsnap version 2015-09-29 with the following settings: –suboptimal-levels = 5 -n 1 -Q –nofails, and providing splice sites from the combined reference file (excluding de novo lincRNAs). Reads were counted for features using htseq-count (mode = union), and using the Ensembl/Necsulea/Guttman reference file (but not including cufflinks-defined novel noncoding RNAs). In total, 672 datasets were analysed, corresponding to 2 × 96 mESCs in 2i, 2 × 96 mESCs in serum + LIF, 96 NPCs (day 6), and 2 × 96 NPCs (day 8).

A number of features were then used to filter out low-quality cells.

(i)  Visual inspection of the IFCs under the microscope (doublets, empty wells, small cells, triplets, burst, dead, debris, and multiple cells).
(ii)  Low-quality cells defined by the cellity package.
(iii)  Cells with <0.5 million mapped reads.
(iv)  Cells with >10% of reads from mitochondrial genes.

For principle component analysis (PCA), only mRNAs and lincRNAs were retained. The cells were normalised using the estimateSizeFactorsForMatrix function from DESeq2, and then PCA performed using the prcomp function in R. The 300 genes making the greatest contribution to each of the first seven principle components were then recorded as the "most variable genes," and these were used for subsequent clustering analyses based on pairwise Euclidean distances. These distances were also used for examining correlated expression of genes in *cis*. To compare

single-cell and bulk RNA-seq data in the same PCA, single cell versus bulk batch effects were first removed using the ComBat package in R.

To measure gene expression heterogeneity between mESCs or between NPCs, we calculated the distance to the median, based on the method described in (Kolodziejczyk et al, 2015). For this, we first calculated the squared coefficient of variation of normalised read counts for each gene, then determined the distance between this measure and a running median (from a scatter plot of squared coefficient of variation values versus mean expression for all genes of a similar length). In parallel for single-cell analysis alone, we preprocessed the data using "Scater" package and performed zero-adjusted normalisation (Lun et al, 2016; McCarthy et al, 2017). We classified highly variable genes with FDR ≤ 0.05 and identified ~3,600 genes, including 113 lincRNAs. Highly correlated lincRNA transcripts and mRNA genes were selected based on correlation coefficient and FDR ≤ 0.05. Representative lincRNA and mRNA genes are shown across a tSNE projection (perplexity = 25) using Rtsne. Cell cycle stages were assigned using a panel of G2M marker genes (Kolodziejczyk et al, 2015).

### CRAC and NET-seq sequencing read preprocessing and filtering

Sequencing reads were preprocessed for standard analyses as follows.

(i) The 3′ adapter was removed (if present) using fastx_clipper.
(ii) Low-quality sequence was removed using the FASTX-Toolkit, specifically fastq_quality_trimmer -t 25, fastq_quality_filter -q 20 -p 90, and fastx_artifacts_filter.
(iii) Reads with identical sequence and 5′ inline unique molecule identifier, UMI, were collapsed, as they are likely to arise from duplication during PCR.
(iv) For CRAC, low-complexity regions were trimmed from the 3′ end of reads (regions of 2 nt or more where 80% or more of the sequence comprises the same nucleotide, for example, … AGAAATAAAAA). This is because many Mtr4-bound RNA fragments contain non-genome-encoded oligo(A) tails that can lead to apparently unique mapping of repeat RNAs.
(v) Homopolymers and dinucleotide-rich sequences were removed using prinseq-lite, with the following settings -min_len 18 -lc_threshold 20 -lc_method dust.

To obtain a set of reads that do not map to repetitive regions of the genome or small noncoding RNAs (e.g., retrotransposons, tRNAs, snoRNAs, and rRNA), we then filtered reads as follows. The reads were mapped separately, using NovoAlign, to three indexes: (i) the mm10 genome, (ii) mm10 protein-coding transcripts (cDNAs), and (iii) pseudochromosomes assembled from RepeatMasker repeats, the ribosomal DNA repeat (Grozdanov et al, 2003) and gencode.vM9 small noncoding RNAs (e.g., miRNA, snoRNA, snRNA, and rRNA). "Nonrepeat" reads were then defined as those that mapped better to the mm10 genome or cDNAs than to any of the repeat RNAs or small noncoding RNAs.

To examine reads arising from Mtr4-bound RNA fragments that contain non-genome-encoded oligonucleotide tails, only those reads for which a 3′ adapter (7 nt or longer) could be detected using

Trimmomatic were retained. The reads were then filtered to retain only those for which each position has a FASTQ quality score of at least 30, and PRINSEQ was used to filter out homopolymers or dinucleotide-rich sequences. Identical sequences (including the inline 5′ UMI) were then collapsed to remove PCR duplicates.

### Identifying reads with non-genome-encoded oligonucleotide tails

Reads for which 3′ adapters were detected and removed (described in the CRAC and NET-seq sequencing read preprocessing and filtering section) were mapped to mm10 using BLASTn (version: ncbi-blast-2.5.0). Any non-mapping region at the 3′ end of the read was recorded as a non-genome-encoded tail, removed from the 3′ end of the read, and annotated at the end of the read name. To exclude reads mapping to repeats, reads were processed as described above, resulting in a set of "nonrepeat" reads. As some of the non-genome-encoded tails might in fact arise from spliced RNA fragments, any reads mapping better to spliced mRNAs than to the genome were also excluded.

To examine the genomic sequence composition around the 3′ end of RNA fragments with non-genome-encoded oligo(A) tails, reads with $A_{\geq 2}$ tails were selected, the genome encoded portion mapped to the genome using bowtie2, and the 3′ end position recorded. Reads were then filtered depending on whether they mapped to introns, PROMPT regions (the 1-kb region downstream of a lincRNA or mRNA TSS, in the sense direction), or antisense PROMPT regions (the 1-kb upstream of a lincRNA or mRNA TSS, in the antisense direction). For genes where more than one A-tailed read mapped to a region, one read was selected at random. The genomic sequence was then extracted for 201-nt windows centred on each of the selected 3′ end positions, and the average nucleotide composition calculated at each position along the PROMPT, antisense PROMPT, and intronic windows. To distinguish intronic A-tailed reads aligning precisely to the 3′ end of the intron from those mapping elsewhere in the intron, we split the intronic reads into those ending with AG versus those that do not.

### CRAC and NET-seq metaplots

For pre-mRNA metaplots, "nonrepeat" reads were mapped to the genome (mm10) using bowtie2 (mode: -sensitive). The midpoint of each mapped read was taken and counted for 5-nt bins flanking each end of exons from protein-coding genes. For each gene, the Appris principal 1 isoform was used (selecting one at random where >1 existed for a single gene). For each transcript, bin totals were divided by the maximum bin total, and then bin averages calculated for each bin across all transcripts. For metaplots of 2-kb regions centred on protein-coding gene TSSes, the same procedure was used, except the bins were 10-nt wide.

### Calculating Mtr4:PolII ratios for PROMPTs, exons, and introns

For this analysis, the set of lincRNA and mRNA annotations including de novo defined Cufflinks features was used. For each transcript, the primary TSS was identified using the following criteria.

(i) The TSS is within 0.5 kb of an H3K4me3 ChIP-seq peak defined by MACS2.

(ii) The sense PROMPT region for the TSS has a moderate NET-seq signal (20 counts).

(iii) The sense PROMPT region has higher NET-seq signal than other TSSes of the same gene.

(iv) There are no other 1-kb regions in the same gene that are not annotated as a TSS but that have > twofold more NET-seq signal than the 1-kb PROMPT region (this criterion excludes genes where the "real" TSS might not be annotated).

(v) The RNA-seq signal in the 2-kb sense direction from the TSS must be at least 1.5× that of the RNA-seq signal in the up-stream 2-kb antisense region (to remove transcripts that are upstream antisense PROMPTs).

(vi) The transcript must not overlap snoRNAs.

(vii) The transcript must be >2 kb, with at least 1 kb of exonic sequence.

For the transcripts corresponding to the selected TSSes, the transcript region was divided into the following categories:

(i) sense PROMPT region: 1-kb downstream of the TSS, sense direction,

(ii) antisense PROMPT region: 1-kb upstream of the TSS, antisense direction,

(iii) exons: all exonic regions outside of the sense PROMPT region, and

(iv) introns.

CRAC and NET-seq reads that do not map to repeats (see the CRAC and NET-seq sequencing read preprocessing and filtering section) were then counted for each of these features, for each selected lincRNA and mRNA transcript, and the Mtr4:PolII ratio calculated.

### Bulk RNA-seq analysis (ribozyme and deletion cell lines, and half-life analysis)

Sequencing reads were aligned to the genome using gsnap, providing splice sites from the combined reference file (excluding de novo lincRNAs) and only retaining uniquely mapping reads. The reads were then counted either for Ensembl features or for the combined reference file (including de novo lincRNAs), using htseq-count (mode = union). For PCA, datasets were normalised using the estimateSizeFactorsForMatrix function in DESeq2, and batches of samples sequenced at different times corrected by centring the means in log space. For analysis of gene expression adjacent to lincRNA loci, the datasets were similarly normalised and batch-corrected by centring the median in log space. Expression changes were evaluated for significance using a Monte Carlo simulation. For transcriptome-wide differential expression analysis, the Wald test in DESeq2 was used (padj < 0.01), including the sequencing data as a factor in the design matrix. To identify genes changing in ex-pression in the Macfarlan *G9a* knock out dataset, *G9a* knock-out and wild-type RNA-seq data were aligned with gsnap, reads counted for Ensembl features, datasets normalised using the estimateSizeFactorsForMatrix function in DESeq2, and the fold change calculated (adding a pseudocount of eight to each mea-surement). Genes with a fold change of >1.5 were recorded as up-regulated. The significance of the overlap between up-regulated

genes in our dataset and the Macfarlan dataset was assessed using a chi-squared test.

For RNA half-life analysis, counts for lincRNAs and mRNAs in the combined reference file (including de novo lincRNAs) were used. Datasets were normalised using size factors calculated for the ERCC RNA spikes (mean count > 50) using the estimateSizeFactorsForMatrix method in DESeq2. To calculate half-lives, the following formula based on median counts at time = 0 and time = 240 was used: $t_{1/2}$ = (240 × log (2))/log(med.ncounts.0/med.ncounts.240).

## Data Availability

All sequencing data have been deposited in the Gene Expression Omnibus, accession code: GSE107493.

## Supplementary Information

## Acknowledgments

We are grateful to Anne Ferguson-Smith for sharing mESC lines. We would like to thank the Friedrich Miescher Institute for Biomedical Research Functional Genomics and FACS facilities, and the Wellcome Trust Sanger Institute Cellular Generation and Phenotyping (cGAP) facility and sample management team, for support with experiments and next-generation se-quencing. This work was supported by the Wellcome Trust [WT103977], [WT206194]; the Swiss National Science Foundation National Centres of Competence in Research RNA & Disease [141735]; and the Friedrich Miescher Institute for Biomedical Research, which is supported by the Novartis Re-search Foundation. K Natarajan was supported by a Wellcome Trust Grant "Tracing early mammalian lineage decisions by single cell genomics" [105031/B/14/Z] and core funding from University of Southern Denmark, Denmark.

### Author Contributions

AC Tuck: conceptualization, resources, data curation, software, formal analysis, supervision, funding acquisition, validation, in-vestigation, visualization, methodology, project administration, and writing—original draft, review, and editing.

K Natarajan: software, formal analysis, investigation, methodology, and writing—review and editing.

GM Rice: conceptualization, validation, investigation, visualization, methodology, and writing—review and editing.

JA Borawski: investigation and methodology.

F Mohn: investigation, methodology, and writing—review and editing.

A Rankova: investigation and methodology.

M Flemr: resources, investigation, and methodology.

A Wenger: resources, investigation, and methodology.

R Nutiu: supervision and methodology.

S Teichmann: conceptualization, resources, supervision, funding acquisition, project administration, and writing—review and editing.

M Bühler: conceptualization, resources, data curation, software, formal analysis, supervision, funding acquisition, validation, investigation, visualization, methodology, project administration, and writing—original draft, review, and editing.

### Conflict of Interest Statement

The authors declare that they have no conflict of interest.

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
