## [Reviewer comments · Life Science Alliance]

Distinctive features of lincRNA gene expression suggest widespread RNA-independent functions

Alex C Tuck, Kedar Natarajan, Gregory M Rice, Jason Borawski, Fabio Mohn, Aneliya Rankova, Matyas Flemr, Alice Wenger, Razvan Nutiu, Sarah Teichmann and Marc Bühler
DOI: 10.26508/lsa.201800124

Review timeline:	Submission Date:	10 July 2018
	Editorial Decision:	10 July 2018
	Revision Received:	12 July 2018
	Editorial Decision:	13 July 2018
	Accepted:	13 July 2018

Report:

(Note: Letters and reports are not edited. The original formatting of letters and referee reports may not be reflected in this compilation.)

Please note that the manuscript was previously reviewed at another journal and the reports were taken into account in inviting a revision for publication at *Life Science Alliance* prior to submission to *Life Science Alliance*.

1st Editorial Decision

10 July 2018

Thank you for transferring your manuscript entitled "An integrative approach distinguishes various activities of mouse large intervening non-coding RNA genes." to Life Science Alliance. The manuscript was assessed by expert reviewers at a different journal before, and their comments were transferred to us by the editor. Based on these reports, we would like to invite you to submit a revised version of your work for publication in Life Science Alliance.

Both reviewers give constructive input on how to revise your analyses with the data already at hand and with re-wording/down-toning your conclusions, and we'd expect and appreciate such a revision. Both reviewers also make good suggestions on how to further develop/provide better support for the ribozyme-based approach. Addressing this issue is not required for publication here.

Thank you for this interesting contribution to Life Science Alliance. We are looking forward to receiving your revised manuscript.

REFeree REPORTS OBTAINED DURING PEER REVIEW ELSEWHERE

Referee #1:

Review of Tuck et al

The biological significance of mammalian lincRNAs remains an unresolved issue. Important questions are whether lincRNAs have any genuine adaptive function, how many are functional/non-functional noise, whether they act in cis or in trans, whether they function as mature RNAs. More technically, methods for loss of function are still evolving, although CRISPR-Cas9 has made a large advance recently.

In this context Tuck et al present a wide-ranging study to address these questions. Using mouse embryonic stem cells to neural precursor differentiation as a model, they seek to address the questions outlined above, by performing loss of function (LOF) of a handful of lincRNAs using

distinct approaches aimed to distinguish cis vs trans regulation and RNA vs DNA functions. They make use of several novel techniques, notably single-cell analysis and knock in of transgenic ribozymes as a means for loss of function.

Sadly, despite the importance of the questions addressed, the paper suffers from serious flaws and one is not convinced that it adds much to the debate. In several aspects its core reasoning is flawed; literature citation is rather disappointing and omits papers that contradict the views of the authors; the ribozyme technology is clever but less effective than what is presently available; technically it is not clear what is the rationale for combining the diverse techniques of single cell RNAseq and ribozyme knockdown and RNA stability. The authors tend to make strong factual statements based on small samples and indirect evidence. This paper does not take us much beyond the general view of lincRNAs: that some are functional, others are difficult to find function for, their genes tend to have features of selected function, although less so than for protein-coding genes, and may give more subtle phenotypes that have arisen more recently in evolution.

MAJOR COMMENTS

Cell model: The authors make extensive use of embryonic stem cells as a model, and seek to draw general conclusions from them. However it is not clear how valid this is. A good example is the Meteor lincRNA, which appears to be non-functional in ESCs (Engreitz 27783602) but has clear roles during mesendoderm differentiation (Micheletti 29180618). Therefore one is not convinced that it is possible to make such general comments about lincRNA functions, which this paper is attempting, based on this ESC model. In other words, a lack of phenotype or function in ESC by no means can be extrapolated more generally.

Transcriptome assembly: The entire paper is based on Cufflinks transcriptome assemblies. The lincRNA field is now aware that such transcript models are highly suspect: almost always incomplete, with many false positive transcripts and many completely lacking gene loci (Steijger 24185837, Hardwick 27502218). While obviously the authors must use some method (StringTie would be better, but still unsatisfactory) they might still comment on this and explore how errors in assembly will affect the paper's conclusions.

Effectiveness of LOF methods: The authors present a nice idea for lincRNA LOF using ribozymes. Unfortunately the method does not work very well: It gives weak knockdown and weak rescue, and it is highly laborious (Figure 4A). Nevertheless the authors repeatedly misrepresent the scientific literature to imply that previous LOF methods (siRNA, shRNA, Gapmer/ASO, CRISPRi, CRISPR-del) are ineffective. Eg Introduction: "direct targeting approaches... can have off target effects and vary in efficacy" - Gapmers tend to be highly effective (there are numerous recent lincRNA papers using them, eg SAMMSON or Wisper papers), and the use of two or more independent Gapmers eliminates most doubts over off-targeting. Similar arguments apply to CRISPRi. Most of these methods are more potent than the ribozyme approach shown here. Another example Results "current approaches...cannot inform whether the RNA itself is functional, or whether the DNA locus or its transcription is more important" - again, Gapmers CAN distinguish these two possibilities. Another: Results "useful addition to the limited range of methods" - siRNA, shRNA, Gapmer, CRISPRi, CRISPR-deletion, CRISPR-RNA is not such a limited range. While there are well known problems, particularly with RNAi methods, it is simply not true that the other methods are not useful solutions for lincRNA LOF experiments. Realistically one does not see the ribozyme method supplanting the others, given their relative effectiveness and convenience. The authors should be more open about the advantages (reversible, stable) and drawbacks (low effectiveness, laborious) of their new method.

LincRNA deletions: The authors also utilize CRISPR deletions of their candidate lincRNAs, Section "Distinguishing cis and trans...". I can find no mention of how the deletions were performed, or how many, nor any data figures relating to the text and results discussed on the first page of that section, nor rationale for choosing the 10 genes mentioned, therefore one simply cannot judge this data. Since a large part of the conclusions depends on this, it is very important. 140 cell lines are mentioned. Did the authors delete entire lincRNAs? How were off-target effects from deleting enhancers / small RNAs / insulators taken into account?

Distinguishing cis/trans effects, Figure 4D: The results are so noisy (eg linc1503, linc1405) and the

numbers of KO cells so few, that it is difficult to draw any conclusions from this data. In figure 4D, the authors perform extremely noise analysis of ten hand picked lincRNAs in ESC cells, and then state "Extrapolating our results, we predict that up to a third of lincRNA loci potentially influence local gene expression...". I simply do not agree that one can draw any general conclusions from this limited sample in these particular cells.

Correlation analysis: The authors make heavy use of correlation analyses (between lincRNAs and mRNAs) to infer the functions and regulatory interactions of lincRNA. This makes the assumption that correlation reflects causation, and that this is specific for lincRNA-mRNA pairs. Many would argue that this is not valid. Many genes (mRNAs included) correlate during cellular differentiation, but this in no way implies that Gene A is directly regulating or regulated by Gene B. Another example: Results "These broad co-expressed genomic regions suggest that cis-regulation involving lincRNA genes could be widespread" - Most neighboring pairs of protein-coding genes are correlated, but it does not mean they regulate each other. Rather it reflects the similar genomic contexts that nearby genes experience.

RNA Stability analysis: The authors also perform stability analysis. It is difficult to see how this fits together with the rest of the paper. There is quite substantial previous literature on this, eg from Mattick's and Ohler's labs, showing that lincRNAs tend to be less stable than mRNAs, but not drastically so. The authors begin by performing classical stability profiling of lincRNAs and mRNAs. Similar to previous studies, they do not find a substantial difference between the two. Strangely, these results are not shown, but described as "inconclusive" and used as motivation to perform a less direct method based on association with nuclear exosome complex Mtr4. It is not explained how it is concluded that they are "inconclusive", is it just because they do not fit with the hypothesis that lincRNAs are less stable? It would be helpful if the authors could show a global correlation analysis between Mtr4 binding and measured half life. It is not justified why the Mtr4 approach is better than convention half life measurements. Overall, even if lincRNAs have different stability profiles than mRNAs, one could interpret these data to simply reflect the fact that most lincRNAs are more recently evolved than mRNAs (eg Ulitsky's work) and therefore have had less time to optimise their gene characteristics for RNA production. The authors end by stating "The predominant output of lincRNA genes is therefore short, promoter-proximal RNAs and not full length transcripts" - while this is certainly the implication of the arguments from this data, I cannot find a direct observation showing it, and therefore such powerfully stated conclusions are not justified.

Off-targeting and repetitive sequences: Some lincRNAs are highly repetitive (including candidate Platr7). Perhaps I missed it, but how do the LOF methods (deletion, ribozyme insertion) cope with this? How can the authors rule out the possibility of off-target ribozyme insertion?

MINOR COMMENTS

It is very difficult to review manuscripts without page numbers.

Units: The authors should use standard units of gene expression being FPKM or TPM.

Missing citations: Abudayyeh, Cox papers.

Low expression and function: At several points in the paper, the authors make the assumption that low expression indicates non-functionality. However this is not a given, and recent papers such as Seiler (28160600) would call this into question.

Introduction: "cases where the RNA product is superfluous" There are indeed cases such as this, but they presently are a small minority compared to numerous instances where the RNA is functional.

Introduction: "LincRNAs are highly differentially expressed between cell types (Cabili et al., 2011), suggesting that lincRNA genes contribute to specification of cell identity and that the RNA product is important" The logic behind this is not clear. Noise can be cell type specific, and I cannot see how one can infer that RNA product is important because of cell type specific expression.

Tuna lincRNA - probably protein-coding. Check latest annotation of Gencode and PhyloCSF tracks in UCSC Browser.

Results: "One notable example...confirming that our approach can help identify regulators of cell identity" - One example from a long list does not confirm the accuracy of the approach.

Referee #2:

In the manuscript "An integrative approach distinguishes various activities of mouse large intervening non-coding RNA genes", Tuck and colleagues investigate the mechanism of action of a subset of lincRNAs in mouse embryonic stem cells. In particular 1) they investigate their expression pattern in individual cells upon neuronal differentiation, comparing bulk and single cell RNA-Seq data; 2) they attempt to dissociate the function of the genomic loci where the lincRNAs are transcribed from the action of the RNA itself, by introducing a new ribozyme-based approach to investigate this question; 3) they further elucidate how lincRNAs are terminated, by investigating the RNA interactome of the exosome factor Mtr4.

While this it is a comprehensive study that is important for the field, it is not totally novel, reiterating recent findings in the literature (for instance, Woo Cho. et al, 2018, Schlackow et al., 2017, Engreitz et al., 2016). It has 3 main parts which are very interesting but that are not explored in enough detail, which would be essential for publication. This is particularly relevant for the ribozyme-based approach, which would need further validation with a higher number of lincRNAs, further optimization for better efficiency and direct comparison with targeting with competing technologies such as ASOs or CRISPRi (not only KO). Only after such evaluation would the lincRNA community be interested in implementing this method, which is indeed promising. In essence, the current version of the manuscript is too broad and the analysis is superficial and does not go into enough detail in many of its parts.

Other points:

- 1) As the authors mention, lincRNAs are lower expressed than mRNAs, which might make them more susceptible to be dropouts in some of the cells. How do the authors distinguish the non-"jackpot" cells from dropouts? The authors should discuss this further.
- 2) The authors state "LincRNAs such as Pvt1 that accumulate to high levels in "jackpot cells" are strong candidates to function as transcripts." Is this really the case? That would exclude lincRNAs as Neat1, Malat1 or 7SK as functional RNAs, which is clearly not the case.
- 3) Pvt1 was recent shown by Howard Chang and colleagues (Woo Cho. et al, 2018) to function through its promoter, via competition with the Myc promoter for enhancer contacts. The authors should mention this paper in the discussion.
- 4) The authors exclude any lincRNAs in the proximity of protein coding genes (5kb). This strategy leads to the exclusion of lincRNAs antisense to transcription factors involved in development (which are likely to be relevant for neural differentiation) or positionally conserved lincRNAs (Amaral et al., Genome Biology 2018). While I understand the authors use this criteria for stringency, they should comment on the fact that relevant lincRNAs might be lost.
- 5) The Supplementary Tables should be formatted in a way that they can be explored in an easier manner. As in example, in Table S3, it would be good to add the gene name along with the ENSEMBL ID.
- 6) The authors should provide a supplementary table identifying the subset of selected lincRNAs that are specifically enriched in the single cell clusters they identified and also the 115 lincRNAs used in Figure 2A.
- 7) The authors should mention which sequencing method is being used for the single-cell RNA-seq, the first time is mentioned
- 8) The authors mention Rmst as a lincRNA associated with the cell cycle. Rmst has been shown to regulate neurogenesis through interaction with Sox2 (Ng et al., Molecular Cell 2013). It would be interesting for the authors to discuss their results in light of this previous literature.

9) The authors compare cell-to-cell variability in their single cell analysis on the neural differentiation in ES cells to clone-to-clone variability in bulk RNA-seq data from 140 cell lines. This variability is most likely derived from distinct biological processes and for me this comparison is not relevant. I would advise the authors to remove this analysis from the paper.

10) The authors state "We next looked for local effects in the neighbourhood of deleted lincRNA loci (Figure 4D), finding that deletion of the linc1405 and Firre genes decreased the expression of the neighbouring genes Eomes and Stk26, respectively." However, it is clear from Figure 4D that this happens only for 3/6 of the deletion cell lines for linc1405 and 3/4 for Firre. Even if it is statistically significant, are the authors confident in that there is a decrease in the case of linc1405? The authors mention buffering effects, could they elaborate more? A strength of the study is indeed the use of a high number of clonal lines for the KD analysis, but what threshold can one be confident with on expression regulation upon KD, when several of the clones do not show such a difference?

11) The authors used for a subset of the experiments male mouse embryonic stem cells from C57BL/6J female × Mus musculus castaneus male (BC8) and Mus musculus castaneus female × C57BL/6J male (CB9) crosses, which allow allelic discrimination. Ideally the HHZ experiments would have been performed in these cell lines, to investigate cis/trans interactions. Since they were not, but only in the ES differentiation experiments, it would be interesting if the authors would have analysed whether the correlation of expression of the lincRNAs and other genes is allele specific, which could give more strength to trans modulation.

12) The knockdown experiments were done in ES cells only, while the function of some of these lincRNAs might be upon differentiation. It would have optimal to perform the experiments in the paper also in neuronal differentiation conditions. While this might be beyond the scope for the current paper, these experiments could possibly reveal cis and trans targets not identified in the current experiments, since the chromatin architecture upon differentiation would be distinct from ES cells.

13) Miat deletion leads to Sox6 and Itrp2. The authors link Sox6 to cardiac muscle development, but it is also involved in neural specification. Could its deletion accelerate specification into specific neural lineages when exposing ES cells to neural differentiation conditions?

14) The authors use a positive control for their ribozyme-based methods the protein coding mRNA for G9a.

a. It was the percentage of overlap of regulated genes between the Macfarlan dataset and HHZ KD? It is not evident from Fig. 3D, which only shows the common upregulated genes.

b. It would be better that the authors would use Miat, which they show acts in trans, or another known trans-acting lincRNA to validate their method (Fig. 4d)

15) Since the ribozyme-based method requires integration of the ribozyme cassette in the locus of the lincRNA of interest, it might be more problematic to implement in lineage restricted cells not as amenable for CRISPR-mediated editing as ES cells. The authors should comment on this.

16) The integration of the ribozyme motif in the genome might disrupt regulatory regions. As such, can the authors rule out that they are not affecting the genomic locus and putative enhancer/promoter functions?

17) The authors claim that the ribozyme approach does not have off-target effects, but they should present data on this, namely by comparing with other technologies as siRNA that have this problem.

18) In Figure 5a, the names of the lincRNAs should be superimposed in the scatter plot, and the yellow dots mentioned in the text are missing.

19) More examples of lincRNAs with early termination should be shown at Fig. 6d.

20) The authors suggest that early termination is a prominent hallmark of lincRNA transcription and that oligoadenylation as a major component of mammalian nuclear decay, but only 8% of the PROMPTs and 10% of Mtr4 RNA bound fragments have oligo A tails. This might be due to the transient nature of interactions of oligo A tails, causing difficulties in capturing such tails. Is this the case in CRAC and orthologue techniques as CLIP? The authors should comment on this.

1st Revision – authors' response

12 July 2018

We thank both reviewers for taking the time to carefully review our manuscript, and for the many suggested improvements. We have now implemented the vast majority of these as outlined in detail below.

A recurring criticism by both reviewers was that our conclusions were too strong based on the data we present. We have therefore toned down our conclusions throughout the manuscript, and highlighted alternative interpretations of the data, for example:

Title: “An integrative approach to dissect mouse lincRNA gene activities” (we refer to the approach, rather than drawing conclusions here)

Abstract: “This suggests that besides RNA-dependent functions, some lincRNA loci act as DNA elements or through transcription.”

Introduction p4: “Reported lincRNA functions include many where the transcript itself is important (e.g. Xist or Fendrr (Chu et al.; Grote et al., 2013)), as well as some cases where the RNA product is superfluous”

Introduction p4: “LincRNAs are highly differentially expressed between cell types (Cabili et al., 2011), and many have shown to help specify cell type by acting as functional RNAs (Guttman et al., 2009; Aelst et al., 2016; Grote et al., 2013; Lin et al., 2014).”

Introduction p6: “...although many lincRNA genes function in trans, we predict that up to a third of lincRNA loci act in cis”

Results p13: “Extrapolating our results, we predict that up to a third of lincRNA loci potentially influence local gene expression, which would be consistent with estimates from two orthogonal studies (Engreitz et al., 2016; Goff et al., 2015). However, this must be tested further, including in different cell types, and ideally testing hundreds or thousands of loci.”

Results p13: “broad co-expressed genomic regions suggest that cis-regulation involving lincRNA genes could be widespread, although this can also reflect co-regulation of nearby mRNA and lincRNA pairs. To distinguish between these possibilities on a gene-by-gene basis will require further lincRNA knock down and deletion experiments.”

Results p13: “However, as no knock down method completely removes the RNA, for loci such as linc1405 that appear to function as DNA elements or via the act of transcription it is impossible to exclude a role for the transcript without additional evidence.”

Results p18: “Together with the local enhancer-like functions of lincRNA loci that we experimentally observed, one possibility is that some of these lincRNA genes act as DNA

elements or through transcription, with the RNA as a dispensable by-product. This is apparently the case for linc1405 and several other lincRNA loci (Engreitz et al., 2016). However, lincRNAs might also act as transcripts despite being low in abundance (Seiler et al., 2017). Therefore, the role of the transcript must always be tested experimentally, taking advantage of the rapidly developing range of lincRNA-directed methods.

Discussion p19: These knock downs did not affect neighbouring gene expression, in contrast to the lincRNA locus deletions, and we speculate that these lincRNA loci act independently of the RNA product.

Discussion p21: The fact that lincRNAs are low in abundance and prone to premature transcription termination and degradation suggests that in some cases the lincRNA gene might not require the RNA product for function. However, some lincRNAs (e.g. VELUCT) function despite having a low abundance (Seiler et al., 2017), and low lincRNA levels could be ideal for some RNA-dependent functions. In summary, it is now clear that when studying a lincRNA gene, it is critical to consider the roles of transcription, the DNA element, and the RNA product (Bassett et al., 2014). This will be greatly aided by the continued development of lincRNA directed techniques, and further exploration of lincRNA properties and their relationship to lincRNA functions.”

Referee #1:

Review of Tuck et al

The biological significance of mammalian lincRNAs remains an unresolved issue. Important questions are whether lincRNAs have any genuine adaptive function, how many are functional/non-functional noise, whether they act in cis or in trans, whether they function as mature RNAs. More technically, methods for loss of function are still evolving, although CRISPR-Cas9 has made a large advance recently.

In this context Tuck et al present a wide-ranging study to address these questions. Using mouse embryonic stem cells to neural precursor differentiation as a model, they seek to address the questions outlined above, by performing loss of function (LOF) of a handful of lincRNAs using distinct approaches aimed to distinguish cis vs trans regulation and RNA vs DNA functions. They make use of several novel techniques, notably single-cell analysis and knock in of transgenic ribozymes as a means for loss of function.

Sadly, despite the importance of the questions addressed, the paper suffers from serious flaws and one is not convinced that it adds much to the debate. In several aspects its core reasoning is flawed; literature citation is rather disappointing and omits papers that contradict the views of the authors; the ribozyme technology is clever but less effective than what is presently available; technically it is not clear what is the rationale for combining the diverse techniques of single cell RNAseq and ribozyme knockdown and RNA stability. The authors tend to make strong factual statements based on small samples and indirect evidence. This paper does not take us much beyond the general view of lincRNAs: that some are functional, others are difficult to find function for, their genes tend to have features of selected function, although less so than for protein-coding genes, and may give more subtle phenotypes that have arisen more recently in evolution.

MAJOR COMMENTS

Cell model: The authors make extensive use of embryonic stem cells as a model, and seek to draw general conclusions from them. However it is not clear how valid this is. A good example is the Meteor lincRNA, which appears to be non-functional in ESCs (Engreitz 27783602) but has clear roles during mesendoderm differentiation (Micheletti 29180618). Therefore one is not convinced that it is possible to make such general comments about lincRNA functions, which this paper is attempting, based on this ESC model. In other words, a lack of phenotype or function in ESC by no means can be extrapolated more generally.

We now include this caveat in the Discussion, using the Meteor lincRNA as an example:

“Some of the lincRNA genes we studied in mESCs may play greater roles in other cell types, as is the case for Meteor (linc1405), which was recently shown to play a role during mesendoderm differentiation (Alexanian et al., 2017).”

One advantage of our ribozyme approach is that, although the knock down is not complete, it is stable, so the effects of lincRNA depletion during differentiation to alternative cell types can be studied.

Transcriptome assembly: The entire paper is based on Cufflinks transcriptome assemblies. The lincRNA

field is now aware that such transcript models are highly suspect: almost always incomplete, with many false positive transcripts and many completely lacking gene loci (Steijger 24185837, Hardwick 27502218). While obviously the authors must use some method (StringTie would be better, but still unsatisfactory) they might still comment on this and explore how errors in assembly will affect the paper's conclusions.

Most of our annotations are from published lincRNA datasets (Ensembl, Guttman et al., 2009, Guttman et al., 2010, Necsulea et al., 2014), and we used Cufflinks/Cuffmerge to supplement these with a de novo assembly to be as comprehensive as possible. All these datasets contain errors in assembly, and so we were very careful to use PolII NET-seq and H3K4me3 ChIP-seq data to filter for “high quality” annotations. We used these where we wanted to be as stringent as possible (the Mtr4 CRAC analyses), and they are included in Table S1 as the “CRAC.XLOCs.HQ/CRAC.biotype” columns. For some analyses (e.g. Figure 6A-C) we compared results using different degrees of filtering, and obtained very similar results.

Many of the lincRNAs that we study in detail (Table 1) have been examined in multiple studies, and therefore the coordinates for these are relatively well established. For those lincRNAs in Table 1 that are not well characterised, we carefully inspected our RNA-seq datasets to ensure the annotations were accurate. We highlight these lincRNAs in green on the transcriptome-wide analyses (e.g. Fig 2B, 2C, 5A, 6A, 6C), to demonstrate that the conclusions we draw from larger, but more error-prone, lincRNA annotations hold true even for a small, but well annotated, set of lincRNAs.

Effectiveness of LOF methods: The authors present a nice idea for lincRNA LOF using ribozymes. Unfortunately the method does not work very well: It gives weak knockdown and weak rescue, and it is highly laborious (Figure 4A). Nevertheless the authors repeatedly misrepresent the scientific literature to imply that previous LOF methods (siRNA, shRNA, Gapmer/ASO, CRISPRi, CRISPR-del) are ineffective. Eg Introduction: "direct

targeting approaches... can have off target effects and vary in efficacy" - Gapmers tend to be highly effective (there are numerous recent lincRNA papers using them, eg SAMMSON or Wisper papers), and the use of two or more independent Gapmers eliminates most doubts over off- targeting. Similar arguments apply to CRISPRi. Most of these methods are more potent than the ribozyme approach shown here. Another example Results " current approaches...cannot inform whether the RNA itself is functional, or whether the DNA locus or its transcription is more important" - again, Gapmers CAN distinguish these two possibilities. Another: Results "useful addition to the limited range of methods" - siRNA, shRNA, Gapmer, CRISPRi, CRISPR-deletion, CRISPR-RNA is not such a limited range. While there are well known problems, particularly with RNAi methods, it is simply not true that the other methods are not useful solutions for lincRNA LOF experiments. Realistically one does not see the ribozyme method supplanting the others, given their relative effectiveness and convenience. The authors should be more open about the advantages (reversible, stable) and drawbacks (low effectiveness, laborious) of their new method.

We have altered the text, and removed some statements, to better acknowledge the power of existing methods, and to discuss the pros/cons of the ribozyme method, for example:

“Approaches to identify functional lincRNA genes are rapidly improving in potency and ease of use. For example, direct RNA targeting approaches using antisense oligonucleotides (ASOs) (Leonelli et al., 2016) or CRISPR-Cas13 (Abudayyeh et al., 2017; Cox et al., 2017) can now efficiently knock down lincRNA expression to directly test the role of the RNA product. This was not the case with previous approaches, such as gene deletion or CRISPR interference, that alter DNA sequence and/or transcription and suffer from unpredictable consequences such as the initiation of novel transcripts (Howe et al., 2017). Despite these recent advances, the efficacy of lincRNA knock downs can be variable, and there may be off target effects. Further improvements to these methods, and the testing of additional strategies, is therefore necessary to obtain rapid, efficient, long-lived and reversible systems to directly target lincRNAs.”

...and...

“Although the ribozyme approach requires some effort to implement and does not always achieve a strong knock down, a major advantage is that the lincRNA depletion is stable and reversible, so the effects of lincRNA depletion at various stages during mESC differentiation to alternative cell types can be studied.”

LincRNA deletions: The authors also utilize CRISPR deletions of their candidate lincRNAs, Section "Distinguishing cis and trans...". I can find no mention of how the deletions were performed, or how many, nor any data figures relating to the text and results discussed on the first page of that section, nor rationale for choosing the 10 genes mentioned, therefore one simply cannot judge this data. Since a large part of the conclusions depends on this, it is very important. 140 cell lines are mentioned. Did the authors delete entire lincRNAs? How were off-target effects from deleting enhancers / small RNAs / insulators taken into account?

We deleted the entire genomic lincRNA loci, cutting with Cas9 as close as possible to the start of the first exon and end of the last exon. We confirmed the deletions by genotyping with two primer pairs, and by RNA-seq, to ensure that they were effective. We now explain the deletions better in the text, and by including an additional Table S4A:

“To examine the functions of our shortlisted lincRNA genes (see Table 1 for main selection criteria), we generated cell lines using CRISPR-Cas9 to delete entire genomic lincRNA loci. A list of cell lines generated, the size of deleted regions, and the Cas9 guides used are provided in Tables S4A and S4B.”

The data from the deletions are shown in Figure S3A, which we now reference more frequently in the text.

In Table 1, we more clearly show the criteria that we used in selecting the lincRNA genes to target (mostly, specific expression in mESCs, according to our RNA-seq data). Notably, many of these lincRNA genes are frequently studied paradigms in the field, and are therefore of widespread interest.

Regarding the question, “How were off-target effects from deleting enhancers / small RNAs / insulators taken into account?” – This is what we were trying to address by comparing the effect of deleting the entire lincRNA locus, to the effect of just depleting the lincRNA transcript. In the case of linc1405, the locus seems to act as an enhancer (also shown by Engreitz et al., 2016) – and could perhaps therefore be redefined as an enhancer rather than a lincRNA. When deleting lincRNA genes, we cut out the minimal region of the genome necessary to remove the transcribed lincRNA exons, without affecting surrounding elements (which might include enhancers and insulators that are not bona fide parts of the lincRNA gene).

Distinguishing cis/trans effects, Figure 4D: The results are so noisy (eg linc1503, linc1405) and the numbers of KO cells so few, that it is difficult to draw any conclusions from this data. In figure 4D, the authors perform extremely noise analysis of ten hand picked lincRNAs in ESC cells, and then state "Extrapolating our results, we predict that up to a third of lincRNA loci potentially influence local gene expression...". I simply do not agree that one can draw any general conclusions from this limited sample in these particular cells.

The data look noisy because when a lincRNA locus is deleted, the effect on a neighbouring gene is often small (e.g. the 2-fold reduction in Eomes expression). Engreitz et al, 2016, saw similar size cis-effects of linc1405 and Bend4, suggesting that these relatively small changes reflect the true biology of lincRNA genes. Importantly, we included sufficient biological replicates to be able to test whether the effects seen are statistically significant (for linc1405 and Firre they are).

We did phrase our prediction that “up to a third of lincRNA loci potentially influence local gene expression” fairly cautiously, and this estimate is consistent with similar studies (Goff et al, 2015, and Engreitz et al, 2016) – which brings the number of lincRNA loci tested in detail to ~30. Nonetheless, this is why we also wanted to approach the problem from a different angle, looking at the behaviour of lincRNA transcripts (stability, transcription, etc).

We have also added a sentence to make it clear that more loci must be tested, before firm conclusions can be drawn:

“Extrapolating our results, we predict that up to a third of lincRNA loci potentially influence local gene expression, which would be consistent with estimates from two orthogonal studies (Engreitz et al., 2016; Goff et al., 2015). However, this must be tested

further, including in different cell types, and ideally testing hundreds or thousands of loci.”

Correlation analysis: The authors make heavy use of correlation analyses (between lincRNAs and mRNAs) to infer the functions and regulatory interactions of lincRNA. This makes the assumption that correlation reflects causation, and that this is specific for lincRNA-mRNA pairs. Many would argue that this is not valid. Many genes (mRNAs included) correlate during cellular differentiation, but this in no way implies that Gene A is directly regulating or regulated by Gene B. Another example: Results "These broad co-expressed genomic regions suggest that cis-regulation involving lincRNA genes could be widespread" - Most neighboring pairs of protein-coding genes are correlated, but it does not mean they regulate each other. Rather it reflects the similar genomic contexts that nearby genes experience.

We have altered the text to discuss both possibilities:

“These broad co-expressed genomic regions suggest that cis-regulation involving lincRNA genes could be widespread, although this can also reflect co-regulation of nearby mRNA and lincRNA pairs. To distinguish between these possibilities on a gene-by-gene basis will require further lincRNA knock down and deletion experiments.”

RNA Stability analysis: The authors also perform stability analysis. It is difficult to see how this fits together with the rest of the paper. There is quite substantial previous literature on this, eg from Mattick's and Ohler's labs, showing that lincRNAs tend to be less stable than mRNAs, but not drastically so. The authors begin by performing classical stability profiling of lincRNAs and mRNAs. Similar to previous studies, they do not find a substantial difference between the two. Strangely, these results are not shown, but described as "inconclusive" and used as motivation to perform a less direct method based on association with nuclear exosome complex Mtr4. It is not explained how it is concluded that they are "inconclusive", is it just because they do not fit with the hypothesis that lincRNAs are less stable? It would be helpful if the authors could show a global correlation analysis between Mtr4 binding and measured half life. It is not justified why the Mtr4 approach is better than convention half life measurements. Overall, even if lincRNAs have different stability profiles than mRNAs, one could interpret these data to simply reflect the fact that most lincRNAs are more recently evolved than mRNAs (eg Ulitsky's work) and therefore have had less time to optimise their gene characteristics for RNA production. The authors end by stating "The predominant output of lincRNA genes is therefore short, promoter-proximal RNAs and not full length transcripts" - while this is certainly the implication of the arguments from this data, I cannot find a direct observation showing it, and therefore such powerfully stated conclusions are not justified.

We have toned down our conclusion, to “In contrast to mRNA genes, lincRNA genes generate more short, promoter-proximal RNAs, and fewer full length transcripts.”

The “inconclusive” results we refer to are the conflicting findings of Melé et al., 2017 and Schlackow et al., 2017, which find that lincRNAs are similarly stable to mRNAs, or less stable than mRNAs, respectively.

We show our own classical stability profiling in Figures 5A and 5B. The differences between lincRNA and mRNA stabilities were small, and not always statistically significant (depending on which set of lincRNA annotations one uses). Therefore, it is not possible to say from this analysis whether lincRNAs are less stable than mRNAs, or not. This is why we developed the Mtr4 CRAC approach.

The Mtr4 CRAC approach is better than conventional half-life measurements because it looks specifically at nuclear RNA decay, and can distinguish at what stage of the RNA life cycle the transcript is degraded, e.g. (i) early transcription termination coupled to RNA degradation, or (ii) degradation of full-length RNAs in the nucleus. Classical half-life measurements do not distinguish between nuclear and cytoplasmic degradation, and certainly cannot tell where/when in the nucleus a transcript is degraded. Furthermore, actinomycin D transcription shut-off is disruptive to the cell, whereas Mtr4 CRAC monitors the endogenous situation.

We show that Mtr4 CRAC data looking at exons (which reports on mature RNA turnover) is consistent with classical half-life data (compare Figure 5B with Figure 6B “exons”) in finding that mature lincRNAs are slightly less stable than mature mRNAs. However, the main conclusion from the CRAC analysis is that lincRNA genes undergo early termination more frequently than mRNA genes do, which is something that cannot be examined using classical half-life analyses.

Off-targeting and repetitive sequences: Some lincRNAs are highly repetitive (including candidate Platr7). Perhaps I missed it, but how do the LOF methods (deletion, ribozyme insertion) cope with this? How can the authors rule out the possibility of off-target ribozyme insertion?

We integrated the ribozymes at non-repetitive sequences, and screened cell lines using two pairs of uniquely mapping PCR primers. We cannot completely rule out the possibility of off-target insertions, but these are very unlikely, as the efficiency of ribozyme integration is fairly low even at the on-target site (5- 25 %), and off-target insertions should be much less frequent. Even if these did occur, they would be at different off-target locations in different biological replicates (of which we have many), so would not generate false positive results.

For the ribozyme integrations, we are confident that there are very few (if any) off-target effects, because we do not see transcriptome-wide changes in gene expression (RNA-seq analysis using DESeq2). We now show an example of this in Figure S3B.

MINOR COMMENTS

It is very difficult to review manuscripts without page numbers.

Sorry – we have now added page numbers.

Units: The authors should use standard units of gene expression being FPKM or TPM.

We show sequencing data as counts, as this is the input for all of our analyses, and the statistical tests we use require count data as an input (e.g. DESeq2 analysis, and single cell analyses). This is particularly important for CRAC data, because each short RNA fragment captured bound to Mtr4 (one “count”) reflects one binding event of Mtr4 to a transcript. Similarly, for NET-seq, one count reflects one elongating polymerase.

Missing citations: Abudayyeh, Cox papers.

We have added these.

Low expression and function: At several points in the paper, the authors make the assumption that low expression indicates non-functionality. However this is not a given, and recent papers such as Seiler (28160600) would call this into question.

We discuss this in the last paragraph of the discussion, and have added the Seiler study as an example.

Introduction: "cases where the RNA product is superfluous" There are indeed cases such as this, but they presently are a small minority compared to numerous instances where the RNA is functional.

We have reworded this sentence:

Reported lncRNA functions include many where the transcript itself is important (e.g. Xist or Fendrr (Chu et al.; Grote et al., 2013)), as well as some cases where the RNA product is superfluous but the act of transcription (e.g. Airn (Latos et al., 2012)) or the underlying DNA element (e.g. Bendl or Lockd (Engreitz et al., 2016; Paralkar et al., 2016)) affects local gene expression.

Introduction: "lincRNAs are highly differentially expressed between cell types (Cabili et al., 2011), suggesting that lincRNA genes contribute to specification of cell identity and that the RNA product is important" The logic behind this is not clear. Noise can be cell type specific, and I cannot see how one can infer that RNA product is important because of cell type specific expression.

We agree that the logic here is not sound, and have changed this sentence to include the pioneering study by Guttman et al. that tested the role of lincRNA transcripts using shRNAs, as well as others that have demonstrated RNA-dependent lincRNA roles in specifying cell type:

"lincRNAs are highly differentially expressed between cell types (Cabili et al., 2011), and many have shown to help specify cell type by acting as functional RNAs (Guttman et al., 2009; Aelst et al., 2016; Grote et al., 2013; Lin et al., 2014)."

Tuna lincRNA - probably protein-coding. Check latest annotation of Gencode and PhyloCSF tracks in UCSC Browser.

Some annotations include a small ORF for Tuna, though it is not clear if this is functional (and if so, the function of the sORF might be to regulate Tuna RNA levels, rather than to generate a functional micropeptide).

Results: "One notable example...confirming that our approach can help identify regulators of cell identity" - One example from a long list does not confirm the accuracy of the approach.

We have removed this section of the manuscript, as also suggested by Reviewer 2.

Referee #2:

In the manuscript "An integrative approach distinguishes various activities of mouse large intervening non-coding RNA genes", Tuck and colleagues investigate the mechanism of action of a subset of lincRNAs in mouse embryonic stem cells. In particular 1) they investigate their expression pattern in individual cells upon neuronal differentiation, comparing bulk and single cell RNA-Seq data; 2) they attempt to dissociate the function of the genomic loci where the lincRNAs are transcribed from the action of the RNA itself, by introducing a new ribozyme-based approach to investigate this question; 3) they further elucidate how lincRNAs are terminated, by investigating the RNA interactome of the exosome factor Mtr4.

While this it is a comprehensive study that is important for the field, it is not totally novel, reiterating recent findings in the literature (for instance, Woo Cho. et al, 2018, Schlackow et al., 2017, Engreitz et al., 2016). It has 3 main parts which are very interesting but that are not explored in enough detail, which would be essential for publication. This is particularly relevant for the ribozyme-based approach, which would need further validation with a higher number of lincRNAs, further optimization for better efficiency and direct comparison with targeting with competing technologies such as ASOs or CRISPRi (not only KO). Only after such evaluation would the lincRNA community be interested in implementing this method, which is indeed promising. In essence, the current version of the manuscript is too broad and the analysis is superficial and does not go into enough detail in many of its parts.

Other points:

1) As the authors mention, lincRNAs are lower expressed than mRNAs, which might make them more susceptible to be dropouts in some of the cells. How do the authors distinguish the non-"jackpot" cells from dropouts? The authors should discuss this further.

This is likely to explain why lincRNAs were originally thought to be more variably expressed between single cells than mRNAs are (which we show is not the case). We were careful to base our conclusions on lincRNAs and mRNAs across all expression levels, as the higher expression levels suffer less from dropout artefacts. We now highlight this in the text:

"We anticipated lincRNAs to be more heterogeneously expressed than mRNAs across single cells (as reported in (Liu et al., 2016)). However, we only observed minor class-wide differences between lincRNAs and mRNAs (Figures 2B and 2C), which held true across all expression levels."

2) The authors state "LincRNAs such as Pvt1 that accumulate to high levels in "jackpot cells" are strong candidates to function as transcripts." Is this really the case? That would exclude lincRNAs as Neat1, Malat1 or 7SK as functional RNAs, which is clearly not the case.

We did not mean to imply that ubiquitously expressed lincRNAs are not likely to be functional. We have therefore changed this sentence, to:

"LincRNAs such as Pvt1 that accumulate to high levels in "jackpot cells" may well function in specific subpopulations of cells."

3) Pvt1 was recent shown by Howard Chang and colleagues (Woo Cho. et al, 2018) to function through its promoter, via competition with the Myc promoter for enhancer contacts. The authors should mention this paper in the discussion.

We have added this paper to the discussion.

4) The authors exclude any lincRNAs in the proximity of protein coding genes (5kb). This strategy leads to the exclusion of lincRNAs antisense to transcription factors involved in development (which are likely to be relevant for neural differentiation) or positionally conserved lincRNAs (Amaral et al., Genome Biology

2018). While I understand the authors use this criteria for stringency, they should comment on the fact that relevant lincRNAs might be lost.

We have added this to the discussion, highlighting how approaches such as ASOs, CRISPR-Cas13 and the ribozyme make it easier to dissect sense:antisense pairs (as these approaches are strand-specific).

5) The Supplementary Tables should be formatted in a way that they can be explored in an easier manner. As in example, in Table S3, it would be good to add the gene name along with the ENSEMBL ID. We have made the Supplementary Tables more accessible (and please note that Table S1 contains a “key” section describing each column in detail, as we are aware that there is a lot of information to navigate), e.g.:

Clearer column name annotations in Table S1A

More friendly column names in Table S2.

Gene names added in Table S3 as requested, and more explanatory column names added.

New Table S4B added with Cas9 guide sequences for gene deletions and size of deleted region. Cas9 guide RNA sequences highlighted in red in Table S5.

Specific columns are now referred to in the text.

6) The authors should provide a supplementary table identifying the subset of selected lincRNAs that are specifically enriched in the single cell clusters they identified and also the 115 lincRNAs used in Figure 2A. We now list all lincRNAs specifically enriched in the Figure 1 single cell clusters in Figure S1.

We have added a list of the lincRNAs used in Figure 2A as Table S1B.

7) The authors should mention which sequencing method is being used for the single-cell RNA-seq, the first time is mentioned

We used SMART-Seq on the Fluidigm C1 System, which we have added to the text.

8) The authors mention Rmst as a lincRNA associated with the cell cycle. Rmst has been shown to regulate neurogenesis through interaction with Sox2 (Ng et al., Molecular Cell 2013). It would be interesting for the authors to discuss their results in light of this previous literature.

We now discuss how the cell cycle specific expression of Rmst might help it fulfil its role as a differentiation factor.

9) The authors compare cell-to-cell variability in their single cell analysis on the neural differentiation in ES cells to clone-to-clone variability in bulk RNA-seq data from 140 cell lines. This variability is most likely derived from distinct biological processes and for me this comparison is not relevant. I would advise the authors to remove this analysis from the paper.

We have removed this analysis.

10) The authors state "We next looked for local effects in the neighbourhood of deleted lincRNA loci (Figure 4D), finding that deletion of the linc1405 and Firre genes decreased the expression of the neighbouring genes Eomes and Stk26, respectively." However, it is clear from Figure 4D that this happens only for 3/6 of the deletion cell lines for linc1405 and 3/4 for Firre. Even if it is statistically significant, are

the authors confident in that there is a decrease in the case of linc1405? The authors mention buffering effects, could they elaborate more? A strength of the study is indeed the use of a high number of clonal lines for the KD analysis, but what threshold can one be confident with on expression regulation upon KD, when several of the clones do not show such a difference?

We are confident that there is a decrease in the case of linc1405, because we include many negative control cell lines (>100) to evaluate whether the effect is significant, and because this same effect is seen by Engreitz et al., 2016 using an independent approach.

We suspect that epigenetic modifications in the Eomes-linc1405 region affect the ability of linc1405 to promote Eomes expression, as the effect seems to vary from clonal cell line to clonal cell line (i.e. must be propagated during cell division). It may be that alternative chromatin regions can compete with linc1405 for contacts to the Eomes locus when appropriately modified, for example.

We now mention this possibility in the text:

"These clonal differences suggest that additional local buffering mechanisms (perhaps epigenetic modifications) maintain neighbouring gene expression and affect the ability of linc1405 to activate Eomes."

11) The authors used for a subset of the experiments male mouse embryonic stem cells from C57BL/6J female × Mus musculus castaneus male (BC8) and Mus musculus castaneus female × C57BL/6J male (CB9) crosses, which allow allelic discrimination. Ideally the HHZ experiments would have been performed in these cell lines, to investigate cis/trans interactions. Since they were not, but only in the ES differentiation experiments, it would be interesting if the authors would have analysed whether the correlation of expression of the lincRNAs and other genes is allele specific, which could give more strength to trans modulation.

This is a very interesting idea. Unfortunately, lincRNA expression was too low for this analysis, as only sequencing reads overlapping with SNPs can be used. However, there are many other allele-specific analyses that the data could, and should, be used for, and we hope that the community will use the data for such analyses once it is released.

12) The knockdown experiments were done in ES cells only, while the function of some of these lincRNAs might be upon differentiation. It would have optimal to perform the experiments in the paper also in neuronal differentiation conditions. While this might be beyond the scope for the current paper, these experiments could possibly reveal cis and trans targets not identified in the current experiments, since the chromatin architecture upon differentiation would be distinct from ES cells.

These experiments would also be very interesting, but we agree that they are beyond the scope of the current manuscript. However, we hope that we, and others, will be able to follow up on the NPC data in the future.

13) Miat deletion leads to Sox6 and Itrp2. The authors link Sox6 to cardiac muscle development, but it is also involved in neural specification. Could its deletion accelerate specification into specific neural lineages when exposing ES cells to neural differentiation conditions?

Along the same lines as for (12), this would indeed be very interesting to follow up on, but would require significant experimental work.

14) The authors use a positive control for their ribozyme-based methods the protein coding mRNA for G9a.

a. It was the percentage of overlap of regulated genes between the Macfarlan dataset and HHrz KD? It is not evident from Fig. 3D, which only shows the common upregulated genes.

We now show the overlap between upregulated genes in our dataset and the upregulated genes in the Macfarlan dataset (highlighted in green).

b. It would be better that the authors would use Miat, which they show acts in trans, or another known trans-acting lincRNA to validate their method (Fig. 4d)

We agree that a trans-acting lincRNA would have been best, which is why we tried Firre and Tuna, but unfortunately did not achieve strong enough knock downs with the ribozyme. Miat is challenging to work with, because it is only low expressed in mESCs, and thus ribozyme knock down experiments would probably need to be performed in NPCs (having first generated the cell lines in mESCs).

15) Since the ribozyme-based method requires integration of the ribozyme cassette in the locus of the lincRNA of interest, it might be more problematic to implement in lineage restricted cells not as amenable for CRISPR-mediated editing as ES cells. The authors should comment on this.

We now mention in the discussion that two advantages of the ribozyme method are (i) that ribozyme mESCs can be differentiated into other cell types to look at the functions of the targeted lincRNAs, and (ii) that it is reversible (so could potentially be used to express a lincRNA at will at particular times during a differentiation).

16) The integration of the ribozyme motif in the genome might disrupt regulatory regions. As such, can the authors rule out that they are not affecting the genomic locus and putative enhancer/promoter functions?

In cases where the ribozyme has an effect on the function of the lincRNA locus, it would indeed be important to check that. For this, an inactive single point mutant ribozyme can be used as a control, which we demonstrate in Figure 3A. We refer to this in the section, "Ribozymes: new tools to investigate lincRNAs".

17) The authors claim that the ribozyme approach does not have off-target effects, but they should present data on this, namely by comparing with other technologies as siRNA that have this problem.

We are confident that integrating a ribozyme into a lincRNA locus does not have off-target effects, because we do not see changes in global gene expression in these cell lines – we now include an example of this in Figure S3B.

18) In Figure 5a, the names of the lincRNAs should be superimposed in the scatter plot, and the yellow dots mentioned in the text are missing.

We have added lincRNA names to Figure 5A. The yellow points referred to in the text are actually in Figure 6A, and we have corrected this erroneous figure reference.

19) More examples of lincRNAs with early termination should be shown at Fig. 6d.

We now show more examples of lincRNAs with early termination.

20) The authors suggest that early termination is a prominent hallmark of lincRNA transcription and that oligoadenylation as a major component of mammalian nuclear decay, but only 8% of the PROMPTs and 10% of Mtr4 RNA bound fragments have oligo A tails. This might be due to the transient nature of

interactions of oligo A tails, causing difficulties in capturing such tails. Is this the case in CRAC and orthologue techniques as CLIP? The authors should comment on this.

Oligo(A) tails are thought to help in the initial recruitment of Mtr4:exosome to transcripts, by acting as a landing pad. We show this using introns as examples, where Mtr4:exosome is presumably recruited at the 3' end, and we indeed see many oligoadenylated RNA fragments. As this complex starts to degrade the RNA and move 3' to 5', one would not expect degradation intermediates to be A-tailed – unless degradation stalls and another cycle of oligoadenylation is used to re-recruit Mtr4:exosome.

Therefore, the fraction of oligoadenylated RNA fragments that we capture is consistent with the role of A- tails in Mtr4:exosome recruitment rather than progression along the transcript. We see a similar proportion of A-tailed reads in *S. cerevisiae* Mtr4 CRAC datasets (Tuck and Tollervey, 2013).

Thank you for submitting your revised manuscript entitled "Distinctive features of lincRNA gene expression suggest widespread RNA-independent functions". We appreciate the point-by-point response provided and the changes made to the manuscript, and are happy to accept your manuscript in principle for publication in Life Science Alliance. Congratulations on this very nice work!
